# Investigating the Vascular Toxicity Outcomes of the Irreversible Proteasome Inhibitor Carfilzomib

**DOI:** 10.3390/ijms21155185

**Published:** 2020-07-22

**Authors:** Panagiotis Efentakis, Hendrik Doerschmann, Claudius Witzler, Svenja Siemer, Panagiota-Efstathia Nikolaou, Efstathios Kastritis, Roland Stauber, Meletios Athanasios Dimopoulos, Philip Wenzel, Ioanna Andreadou, Evangelos Terpos

**Affiliations:** 1Laboratory of Pharmacology, Faculty of Pharmacy, National and Kapodistrian University of Athens, 15771 Athens, Greece; pefentakis@pharm.uoa.gr (P.E.); nayanik@pharm.uoa.gr (P.-E.N.); 2Cardiology I Department, University Medical Center of the Johannes Gutenberg-University Mainz, 55131 Mainz, Germany; h.doerschmann@uni-mainz.de (H.D.); c.witzler@uni-mainz.de (C.W.); wenzelp@uni-mainz.de (P.W.); 3Molecular and Cellular Oncology/ENT, University Medical Center of the Johannes Gutenberg-University Mainz, Langenbeckstr. 1, 55101 Mainz, Germany; svenja.siemer@uni-mainz.de (S.S.); rstauber@uni-mainz.de (R.S.); 4Department of Clinical Therapeutics, School of Medicine, National and Kapodistrian University of Athens, 11528 Athens, Greece; ekastritis@gmail.com (E.K.); mdimop@med.uoa.gr (M.A.D.); eterpos@med.uoa.gr (E.T.); 5Center for Cardiology—Cardiology I, University Medical Center Mainz, Langenbeckstraße 1, 55101 Mainz, Germany; 6German Center for Cardiovascular Research (DZHK), Partner Site Rhine-Main, Germany

**Keywords:** carfilzomib, vasculature, vascular smooth muscle cells, autophagy, endoplasmatic-reticulum stress

## Abstract

Background: Carfilzomib’s (Cfz) adverse events in myeloma patients include cardiovascular toxicity. Since carfilzomib’s vascular effects are elusive, we investigated the vascular outcomes of carfilzomib and metformin (Met) coadministration. Methods: Mice received: (i) saline; (ii) Cfz; (iii) Met; (iv) Cfz+Met for two consecutive (acute) or six alternate days (subacute protocol). Leucocyte-derived reactive oxygen species (ROS) and serum NO_x_ levels were determined and aortas underwent vascular and molecular analyses. Mechanistic experiments were recapitulated in aged mice who received similar treatment to young animals. Primary murine (prmVSMCs) and aged human aortic smooth muscle cells (HAoSMCs) underwent Cfz, Met and Cfz+Met treatment and viability, metabolic flux and p53-LC3-B expression were measured. Experiments were recapitulated in AngII, CoCl_2_ and high-glucose stimulated HAoSMCs. Results: Acutely, carfilzomib alone led to vascular hypo-contraction and increased ROS release. Subacutely, carfilzomib increased ROS release without vascular manifestations. Cfz+Met increased PGF2α-vasoconstriction and LC3-B-dependent autophagy in both young and aged mice. In vitro, Cfz+Met led to cytotoxicity and autophagy, while Met and Cfz+Met shifted cellular metabolism. Conclusion: Carfilzomib induces a transient vascular impairment and oxidative burst. Cfz+Met increased vascular contractility and synergistically induced autophagy in all settings. Therefore, carfilzomib cannot be accredited for a permanent vascular dysfunction, while Cfz+Met exert vasoprotective potency.

## 1. Introduction

Multiple myeloma (MM) accounts for 1% of neoplastic diseases [1,2], and the irreversible proteasome inhibitor (PI) carfilzomib (Cfz) stands as an important anti-myeloma agent in relapsed/refractory myeloma (RRMM) patients. The ENDEAVOR (Carfilzomib and dexamethasone versus bortezomib and dexamethasone for patients with relapsed or refractory multiple myeloma) study demonstrated a significant survival benefit in RRMM patients who received carfilzomib compared to patients who received the reversible inhibitor bortezomib (Btz) [3]. However, carfilzomib is associated with more frequent occurrence of cardiovascular events, which leads to acute but reversible cardiac failure [1,4].

MM is predominantly a disease of the elderly [5] and the incidence of pretreatment baseline cardiovascular comorbidities increases with age [6]. The presence of comorbidities in MM patients is recognized as an independent risk factor for cardiotoxicity. At the same time, cardiovascular function, biochemical markers and the PI dose do not correlate with the manifestation of cardiomyopathy [3]. The prediction of cardiovascular adverse events remains an unmet clinical need [7,8] and continuous cardiac monitoring of carfilzomib-treated patients remains necessary [7,8]. Data on the prophylactic treatments in patients with carfilzomib-induced cardiotoxicity are also limited [9,10].

The ENDEAVOR study showed that carfilzomib-treated patients have a significantly increased risk of presenting with vascular diseases (i.e., arterial hypertension (14–18%); peripheral edema (24%)) compared to bortezomib-treated patients (3–7%; 1–18% respectively) [1,3,5]. However, the direct effect of carfilzomib on the vasculature is unclear. A translational in vivo model of carfilzomib-induced cardiotoxicity has been recently established [11], but the vascular effects of carfilzomib were not investigated. The purpose of the present study is to investigate the effects of carfilzomib on vascular function in/ex vivo and in vitro. Based on the recent data supporting a protective role for metformin against carfilzomib-induced cardiotoxicity in vivo [11], we further studied the effects of carfilzomib and metformin coadministration on the vascular phenotype. More specifically, we investigated: the (i) acute and (ii) subacute effects of carfilzomib on vascular activity. We set out to identify the effects of metformin coadministration on a. the vascular phenotype in vivo in young and aged mice, on b. young primary murine vascular smooth muscle cells (VSMCs) and c. senescent human aortic smooth muscle cells (HAoSMCs) in the presence and absence of cardiovascular risk stimuli in vitro, focusing on the possible functional and metabolic changes.

## 2. Results

### 2.1. Carfilzomib Led to an Acute Decrease in PGF2α-Induced Aortic Constriction and Leucocyte-Derived Oxidative Burst, Independently of Metformin Cotreatment

To investigate the acute effect of carfilzomib on vascular function, we used the consecutive 2-day carfilzomib treatment protocol with coadministration of metformin treatment [11]. Aortas did not show any signs of vascular dysfunction percentage-wise, as shown by their vasodilatory response to both acetylcholine (ACh) and nitroglycerine (Gtn; Figure 1A,D). Assessment of tension (g) of the aortas in the treated mice, showed that carfilzomib led to a decreased initial contraction response to prostaglandin F2α (PGF2α; Figure 1F) and to a significant aggravation of endothelial (Figure 1B) and smooth muscle cell-mediated vasorelaxation (Figure 1E), an effect that was not ameliorated by metformin. Moreover, carfilzomib led to increased ROS release (oxidative burst) (Figure 1G,H), an effect that was only partially inhibited by metformin in the Cfz+Met group, in terms of cumulative leucocyte ROS release mediated by phorbol 12,13-dibutyrate (PdBu) (Figure 1H). These changes in the ROS releasing capacity by the leucocytes were not associated with changes in the white blood cell count in the whole blood (Figure 1I).

### 2.2. Carfilzomib Subacute Treatment Did not Induce Vascular Dysfunction, while Cotreatment With Metformin Led to an Increased Response to PGF2α

Identifying a possible acute effect of carfilzomib on vascular function, we sought to investigate whether this effect was sustained or compounded in the subacute settings. Our results from the 6-days protocol showed that subacute carfilzomib treatment did not lead to any vascular dysfunction either percentage-wise or in terms of vascular tension (Figure 2A,B,D–F). On the contrary, the coadministration of carfilzomib and metformin led to a statistically significant increase in PGF2α responsiveness (Figure 2F), which establishes an altered endothelial (Figure 2B) and vascular smooth muscle cell-mediated relaxation (Figure 2E). Moreover, subacute carfilzomib administration led to a sustained ROS release (Figure 2G,H), which was partially decreased by metformin both in terms of macrophages and leucocytes oxidative burst (Figure 1G), without a significant change in white blood cell count in whole blood (Figure 2I).

Therefore, the in/ex vivo vascular studies demonstrated an acute transient vascular dysfunction induced by carfilzomib in vivo, which is subsequently resolved. On the contrary, subacute carfilzomib and metformin coadministration led to an increased vascular contraction potential, which was further investigated.

### 2.3. Coadministration of Carfilzomib and Metformin Led to an Enhanced Ampkα-Endoplasmic Reticulum Stress-Dependent Autophagy and a Decreased A-Smooth Muscle Cell Actin Expression in Vivo

We assessed the histological vascular phenotype in the subacute protocol in the context of identifying potential permanent vasculature damage regarding altered wall thickness and collagen deposition. Carfilzomib administration alone did not induce changes in the vascular wall thickness as assessed by hematoxylin–eosin (H&E) staining (Figure 3A (upper panel),B), while Cfz+Met cotreatment led to a significant increase in collagen deposition (Figure 3A (lower panel),C). Immunoblotting analysis showed that metformin decreased mammalian target of rapamycin (mTOR) phosphorylation versus control. Carfilzomib caused a decrease in the phosphorylation of mTOR, AMP-activated kinase α (AMPKα) and protein kinase B (Akt) compared to the control and to increased mTOR, inositol-requiring enzyme-1α (IRE1α), inducible nitric oxide synthase (iNOS), calnexin, binding immunoglobulin protein (BIP) and LC3-B expression, an upregulation of endoplasmic reticulum (ER) stress (as shown by the ER stress markers IRE1α, calnexin and BIP) and increased autophagy (as shown by the autophagy marker LC3-B). Interestingly, coadministration of carfilzomib and metformin led to an even greater decrease in mTOR phosphorylation and αSMA expression, as well as an increase of LC3-B and IRE1α expression compared to the control group, demonstrating sustained ER stress and increased autophagy in the aortas. Coadministration of the drugs led to a minor but significant decrease in BIP and calnexin expression, which remained significantly increased compared to the control group, and to a restoration of both AMPKα and Akt phosphorylation, as demonstrated previously by our group in the murine myocardium [11] (Figure 3D,E). Importantly, endothelial markers such as endothelial nitric oxide synthase (eNOS) and vasodilator-stimulated phosphoprotein (VASP) phosphorylation and expression were not affected by treatments administration. The increased LC3-B and decreased αSMA expression was further confirmed by immunofluorescent confocal microscopy (Appendix A). Conclusively, carfilzomib appeared to induce autophagy and ER stress in the aortic vasculature. Carfilzomib and metformin coadministration amplified the autophagic signaling and decreased vascular αSMA expression, with no effect on the endothelial homeostasis.

### 2.4. Carfilzomib Treatment Did Not Affect Endothelial Homeostasis and Did Not Induce Nitro-Oxidative Stress in the Murine Aortas

To further investigate the endothelial homeostasis as well as the redox status of the carfilzomib- and metformin-treated mice, the total circulating nitrates/nitrites (NO_x_) content in the sera as a biomarker of NO release, 3-nitrotyrosine (3-NT), nicotinamide adenine dinucleotide phosphate (NADPH) oxidase 2 (*nox2*) and NADPH oxidase 4 (*nox4*) redox-related genes in the aortic tissues as markers of nitro-oxidative stress were assessed [12]. Neither carfilzomib nor metformin or the drugs coadministration led to any significant changes in NO_x_ and 3-NT content (Figure 4A,B), whereas *nox2* and *nox4* levels did not differ among groups (Figure 4C). Based on these results, endothelial homeostasis was unaffected by carfilzomib and metformin at the selected doses in the subacute protocol and subacute carfilzomib administration did not lead to nitro-oxidative mediated damage of the aortic tissue.

Subsequently, we investigated the endothelial homeostasis of the treated murine aortas. Using a real-time PCR (RT-PCR) analysis we assessed key proinflammatory and regulatory genes of the endothelial cells, which included intercellular adhesion molecule 1 (*icam1*), vascular adhesion molecule 1 (*vcam1*), endothelial selectin (*sele*) and tissue factor (*tf*). Carfilzomib caused an increase in *icam-1* mRNA levels in the aortas, partially abrogated by metformin administration and an upregulation of tf mRNA levels, which was independent of metformin administration. *Vcam-1* and *sele*, did not differ among study groups. Therefore, carfilzomib might lead to a proinflammatory and prothrombotic phenotype of the vessels, as shown by *icam-1* and *tf*, respectively, which requires further investigation.

### 2.5. Carfilzomib Increases NO_x_ Content in the Serum of the Aged Mice and Upregulates Inos and BIP in the Aortic Tissue. Coadministration of Carfilzomib and Metformin Leads to an Induction of LC3-B-Dependent Autophagy in the Aged Murine Aortas

Given the higher incidence of MM among the elderly population [5], we also assessed the effects of subacute carfilzomib and metformin treatment in an aged murine model. As per the data in young mice, carfilzomib did not induce nitro-oxidative damage in the aortas, as demonstrated by the unchanged levels of 3-NT (Figure 5C). However, aged mice exhibited decreased serum NO_x_ levels, and carfilzomib administration increased circulating NO_x_ (Figure 5A,B), which was associated with an upregulation of iNOS in the aortic tissue. In agreement with the mechanistic data from the young murine aortas, carfilzomib upregulated BIP, a marker of the unfolded protein response. Carfilzomib and the coadministration of carfilzomib and metformin did not affect Raptor phosphorylation or expression but led to an increase in mTOR inhibitory phosphorylation at S2481. Metformin increased eNOS phosphorylation compared to the control group, and metformin as well as the drug coadministration increased both eNOS and AMPKα phosphorylation compared to carfilzomib alone. Furthermore, metformin and Cfz+Met led to increased LC3-B expression (Figure 5D,E). Importantly and in compliance with the young mice data, the drugs’ coadministration led to a decrease in αSMA.

Collectively, the mechanistic data obtained from both the young and the aged murine models show that subacute coadministration of carfilzomib and metformin induces a LC3-B-dependent autophagy in the vessels, which is independent of age. These effects were observed mostly in the vascular smooth muscle cell layer of the aortas.

### 2.6. Carfilzomib and Metformin-Induced Cytotoxicity in Primary Murine Vascular Smooth Muscle Cells Mediated by a Cellular Metabolism Shift and LC3-B Dependent Autophagy

Following our in vivo findings, we performed in vitro experiments on isolated primary vascular smooth muscle cells. Taking under consideration that cytotoxicity of VSMCs, as indicated by decreased VSMCs viability (decreased MTT conversion and confluency), is associated with increased vascular plasticity [13], the effect of the compounds on the viability of prmVSMCs was initially investigated. Carfilzomib, metformin and the combination of the drugs decreased metabolic conversion of MTT, showing signs of cytotoxicity in all selected doses, a finding that is in agreement with the literature (Figure 6A) [14,15]. We selected the doses of 0.3 μΜ and 10 mM for carfilzomib and metformin respectively, as the most potent doses, and experiments were repeated in order to test their effects on cellular confluency as a marker of cellular death. No effect on cellular confluency was observed (Figure 6B). Given that carfilzomib decreased AMPKα phosphorylation and metformin is a well-known AMPKα activator, we performed in vitro metabolic flux analyses to assess the compounds’ metabolic effect. Metabolic flux analyses showed that metformin shifted cellular metabolism and ATP production from oxidative phosphorylation to glycolysis under normal and low glucose conditions (Figure 6C–F). Moreover, while carfilzomib did not shift the cellular metabolism, it led to a decrease in ATP production via oxidative phosphorylation under conditions of both normal (Figure 6C) and low glucose conditions (Figure 6D) compared to the control (Normal Saline; NS 0.9%). Finally, to investigate whether the aforementioned effects led to apoptosis or autophagy in the prmVSMCs we investigated the expression of p53 (a marker of apoptosis) and LC3-B (a marker of autophagy) by confocal and automated immunofluorescent microscopy. Treatment administration led to a significant decrease in p53 expression and an increase in LC3-B signal compared to controls, demonstrating that the induced pathway is autophagy-dependent and apoptosis-independent (Figure 6G–I).

### 2.7. Carfilzomib and Metformin Led to Decreased Cellular Confluency in Senescent Human Aortic Smooth Muscle Cells, while Metformin Shifts Cellular Metabolism to Glycolysis

To increase the translational value of our experiments, we performed experiments on senescent HAoSMCs under the same conditions as in the prmVSMCs. We found that, in contrast to the prmVSMCs, the compounds did not reduce MTT conversion significantly in the HAoSMCs, while metformin at the low dose led to an increased MTT metabolic conversion (Figure 7A). However, carfilzomib and Cfz+Met decreased cellular confluency compared to controls (Figure 7B). Concerning the metabolic flux analysis, metformin led to a total shift of ATP production from oxidative phosphorylation to glycolysis, in accordance with the prmVSMCs, while in this setting carfilzomib did not affect ATP production compared to controls under conditions of both normal and low glucose (Figure 7C–H), indicating that metabolic alterations induced by metformin play a crucial role in the observed phenotype both in metformin and Cfz+Met treated senescent cells.

### 2.8. Carfilzomib and Metformin Induce Cytotoxicity on Senescent HAoSMCs in the Presence of Comorbidities in an LC-3B-Dependent Mechanism

To simulate clinical observations in vitro, we subjected the senescent HAoSMCs to cardiovascular risk stimuli, such as angiotensin II (AngII)-simulating hypertension, CoCl2-simulating hypoxia and high glucose-simulating hyperglycemia. AngII and CoCl2 led to an increased MTT metabolic conversion, as demonstrated previously by other groups (Figure 8A) [16,17,18]. In agreement with findings in the untreated senescent cells, metformin and Cfz+Met led to significant cytotoxicity (Figure 8A) in the context of all comorbidity stimuli. Concerning cellular confluency, carfilzomib and Cfz+Met showed a compliant decrease in all tested settings (Figure 8B). Finally, to investigate whether the effect was apoptosis- or autophagy-mediated we performed immunofluorescent experiments of p53 and LC3-B expression. Confirming earlier reports [15,19,20,21,22,23,24], AngII (100 nM) and glucose (25 mM) led to an increase of p53 expression in HAoSMCs. In compliance with the prmVSMCs, all treatments led to a decrease in p53 expression in the absence of cardiovascular risk stimuli (Figure 8C,D), an effect that was consistent among all comorbidity settings and significant in the high glucose cohort compared to Normal Saline (NS) 0.9% (Figure 8C,D). On the contrary, in all tested conditions, a significant increase in LC3-B expression was observed only in the Cfz+Met group (Figure 8E).

## 3. Discussion

Carfilzomib is a potent antimyeloma agent with an overall favorable toxicity profile, but a consistent and reproducible signal of cardiovascular toxicity has been reported [25]. In this context and given the limited data available in the literature, we sought to shed light on the effects of this agent on the vasculature by assessing vascular contractility and relaxation. This is the first preclinical study investigating the direct effect of carfilzomib on the vasculature in/ex vivo and in vitro, in the absence and presence of cardiovascular comorbidities [5].

Our results showed that carfilzomib leads to an acute, transient vascular hypo-reactivity, and induces endoplasmic reticulum (ER) stress-mediated signaling and AMPKα dephosphorylation in the aortic musculature in the subacute protocol, without any permanent vascular or endothelial damage. In contrast however to the autophagy-inhibitory effect of carfilzomib in the myocardium, carfilzomib appears to induce autophagy in the vessels, in an ER-stress dependent manner, despite the consistent dephosphorylation of AMPKα in both organs [11]. We have previously demonstrated that metformin prevents carfilzomib induced myocardial toxicity in young mice [11] and in that context we aimed to assess whether metformin can also act as a protective agent against carfilzomib induced vascular toxicity [25]. Carfilzomib and metformin coadministration led to the restoration of AMPKα phosphorylation and increased LC-3B-dependent autophagy in an AMPKα/ER-stress dependent pathway, as well as through metabolic changes in SMCs both in vivo and in vitro (Figure 9). These findings indicate that the effect of carfilzomib on the vasculature is mainly vascular smooth muscle cell-dependent and involves the regulation of LC3B-autophagy, which is further amplified by metformin.

Bortezomib, a reversible proteasome inhibitor, has a cytoprotective effect on endothelial cells mediated by the upregulation of antioxidant enzymes through the nuclear factor erythroid 2-related factor 2 (Nrf2)-axis [26]. Bortezomib’s vasoprotective effects are evident in various vascular disease animal models, namely pulmonary hypertension and aortic aneurysms [27,28,29]. On the contrary, data concerning carfilzomib vascular outcomes are ambiguous and vascular effects of PIs seem to be drug- rather than class-dependent.

Carfilzomib inhibits arterial hypertrophy by ubiquitinating p62, the heat shock protein (HSP90) and inducing LC3-B-dependent autophagy-related cell death in hypertrophic SMCs of the rat aorta. This is consistent with our data, which demonstrate that carfilzomib induces SMC-derived autophagy via an LC3-B dependent mechanism, in the absence of vascular diseases [14]. Case studies have however linked carfilzomib treatment to systemic and pulmonary hypertension and renal toxicity, which is contradictory to the in vivo vasoprotective potency of the drug [9]. Moreover, ex vivo experiments have demonstrated a carfilzomib-induced immediate (within 24 h) spasmogenic effect [30,31]. Following these results, we have shown that carfilzomib led to a subsequent temporary decreased responsiveness of the vessel within 48 h, which was resolved after 6 days of treatment. Rapid induction of the oxidative stress, as observed in our in vivo model of acute carfilzomib treatment, has been shown to induce a SMC switch from a contractile to a synthetic (matrix protein excreting) phenotype [32]. This could justify the decreased vasoconstrictive capacity of the aortas, observed following 48 h of carfilzomib treatment. The induction of autophagy, as observed after subacute carfilzomib administration, increased SMCs contractility [33,34] and might explain the resolution of the hypo-contractile phenotype. Therefore, it should be noted that the direct effects of carfilzomib on vascular function seem to be transient and cannot explain the observed clinical vascular presentations [31].

Moreover, carfilzomib did not alter nitric oxide (NO) homeostasis as assessed by circulating NO_x_ levels nor led to nitro-oxidative damage or redox alterations in the aortas, as shown by Figure 3 NT and *nox2-nox4* mRNA levels. On the contrary, carfilzomib increased *icam-1* mRNA levels, revealing a proinflammatory phenotype of the vessels. It has been previously demonstrated that the increased circulating ROS, as observed in our protocol in the Cfz-treated leucocytes (Figure 2G,H), can induce endothelial ICAM-1 expression [35]. Therefore, circulatory oxidative stress can be accredited for *icam-1* mRNA upregulation in Cfz-treated aortas. Histological evaluation of the aortas (Figure 3A) did not however provide any evidence for immune cell infiltration of the aortas. Therefore, the induction of inflammation in the subacute protocol cannot be established. In agreement with the ROS-releasing capacity of the leucocytes isolated by mice treated with the combination of carfilzomib and metformin, the drug’s combination seems to partially abrogate *icam-1*, as a result of the mitigated oxidative stress in the circulation. This mild proinflammatory phenotype might be of interest in a chronic model of carfilzomib treatment, aimed at investigating the effect of the drug on the vascular function in the long term.

Concerning the regulation of TF of the aortas, we found that carfilzomib increased *tf* mRNA levels, independently of the metformin treatment. TF is a key regulatory integral membrane protein on the surface of endothelial cells, implicated in atherosclerosis, and a major procoagulant for thrombus formation associated with endothelial dysregulation [36]. To the best of our knowledge, there are no data on the effect of carfilzomib on TF regulation. However, there are data supporting a prothrombotic activity of carfilzomib, as implicated by the common complication of acute thrombotic microangiopathy (TMA) seen in carfilzomib treated patients [37,38]. It is, therefore, possible that carfilzomib induces a (pro)thrombotic phenotype in the vessels, but investigating this action was not within the scope of the current study. The effect of carfilzomib on the coagulation phenotype and on microvascular thrombosis will be addressed in future research work.

Metformin is a widely used anti-hyperglycemic drug with prophylactic activity against cardiovascular diseases [39]. It has a direct effect on SMCs through AMPKα activation, and also anti-inflammatory activity on SMCs [22]. In vivo, metformin inhibits aortic aneurysm formation, vascular hyperplasia and pulmonary hypertension through activation of AMPKα and improves antioxidant capacity and mitochondrial homeostasis [15,19,20,21,23]. The effect of metformin on autophagy is controversial. It inhibits autophagy in pathological conditions such as metabolic syndrome and pulmonary hypertension through activation of Akt and inhibition of mTOR-C2 [24,40]. The activation of AMPKα by metformin can also induce autophagy by inhibiting the mTOR-C1 complex, which has been confirmed in tumor cells, cardiomyocytes and in our previous in vivo study in the myocardium [11,41,42]. Thus, the effect of metformin on the induction or inhibition of LC-3B-dependent autophagy depends on the disease burden of cells. In our study, metformin enhanced the carfilzomib-induced autophagy observed in the vessels. The increased autophagic signaling noted in the metformin, carfilzomib and combination groups was also demonstrated by the reduced phosphorylation of mTOR, as activatory phosphorylation of mTOR leads to decreased autophagy by inhibiting the ULK1 complex [13]. The two drugs seem to trigger autophagy through different pathways and to lead synergistically to the observed vascular phenotype.

Surprisingly, combination therapy decreased αSMA and increased perivascular fibrosis in the in vivo subacute protocol. Downregulation of αSMA, a marker of SMCs, is supported to decrease vascular stiffness and increase vascular contractility [43]. Induction of autophagy in SMCs is known to decrease αSMA, increase SMC contractility and induce the SMC-dependent secretion of collagen [33,34]. The latter could justify the increased perivascular fibrosis observed in the Cfz+Met group. However, perivascular collagen deposition, which can play both a protective and a dysfunctional role in the vessels [44], needs further investigation.

To increase the translational value of our experiments, we incorporated an aged murine model to investigate the mechanistic effects of carfilzomib on the aged vasculature. In agreement with data from the young mice, carfilzomib did not induce nitro-oxidative damage on the vessels as demonstrated by its surrogate marker 3-NT, despite the increased serum NO_x_ concentration and iNOS expression seen in the carfilzomib-treated aged aortas. Mechanistically, metformin increased AMPKα and eNOS phosphorylation—independently of the carfilzomib treatment—an effect that is already shown to be vasoprotective in aged vessels [45]. Carfilzomib and metformin coadministration led to increased AMPKα and eNOS phosphorylation, as well as to a synergistic induction of autophagy as shown by mTOR inhibitory phosphorylation at S2481, and increased LC3-B expression. Moreover, a significant decrease of αSMA—a surrogate marker of the VSMCs—was observed only in the Cfz+Met group. The induction of autophagy in aged vessels ameliorates vascular hypertrophy and exerts vasoprotective effects, by decreasing αSMA expression as a result of decreased VSMCs proliferation [46]. Therefore, carfilzomib and metformin coadministration exerts a significant vasoprotective effect on the aged vessels via increased vascular LC3-B-dependent autophagy and increased eNOS and AMPKα phosphorylation.

Following our in vivo experiments, which showed that combination therapy leads to a synergistic induction of autophagy in the SMCs, which can be potentially protective in the presence of cardiovascular comorbidities [47], we assessed this effect in primary murine and human smooth muscle cells, to investigate in depth the mechanistic nature of the observed in vivo effects. Carfilzomib and metformin exhibited cytotoxicity on VSMCs through metabolic alterations and autophagy. The latter mechanism is in accordance with our in vivo experiments, where amplified autophagy was also observed. Even though carfilzomib did not lead to a shift in cellular bioenergetics, it led to a decrease in ATP production via oxidative phosphorylation. Carfilzomib is known to induce a mitochondrial deficit in resistant MM cells in vitro in a mitochondria-derived activator of caspase (SMAC)-Survivin dependent manner leading to increased apoptosis, mitochondrial-derived ROS release and inhibition of mitochondrial respiration, which is in line with our data [48,49]. It should be stressed at this point that the metabolic switch induced by metformin might be of great clinical importance, as drugs that can modulate cellular bioenergetics in a similar manner, are being successfully incorporated in the management of heart failure [50]. To increase the translational value of the study, we also set up an in vitro model of aging (senescent HAoSMCs) [15]. We observed that metformin reduced cell viability only in the presence of comorbidities. This finding suggests that carfilzomib largely loses its effectiveness in aging cells, possibly due to a reduction in the activity of the proteasome [51] while metformin retains its anti-proliferative properties in accordance with previously published in vitro data [15]. Additionally, the combination therapy decreased p53 and increased LC3B expression under all experimental conditions. Activatory phosphorylation of p53 has been linked to induction of apoptosis, whereas p53 ubiquitinylation leads to proteasomal destruction of the protein [52]. Given the decreased expression of p53, the anti-proliferative effect observed seems to be apoptosis-independent and LC-3B-autophagy-dependent. Conclusively, the induction of autophagy in vitro and in vivo in the coadministration group is responsible for reducing cell viability, modifying cellular phenotype/metabolism and increasing vascular plasticity. The decreased VSMC viability and increased LC-3B-dependent autophagy is proven to protect against vascular diseases, as it reverses vascular stiffness and hypertrophy [13].

The current findings are of great clinical importance given the significant impact of acute heart failure associated with carfilzomib administration. The increased vascular plasticity observed when carfilzomib is coadministered with metformin may offset this adverse effect by decreasing cardiac afterload [53]. Additionally, carfilzomib/metformin coadministration may ameliorate smooth muscle cell hypertrophy and vascular stiffness, improving the vascular phenotype and acting as a pleiotropic prophylaxis in MM patients [47]. It must be pointed out that carfilzomib does not seem to affect endothelial homeostasis and therefore does not induce endothelial dysfunction. Research efforts to manage carfilzomib induced vascular deficits should, therefore, focus on the vascular smooth muscle cell compartment of the vessels. Thus, the establishment of carfilzomib/metformin combination therapy as an appropriate regimen for maintaining cardiovascular homeostasis, particularly in MM patients with a high cardiovascular comorbidity burden, may contribute to an improvement in the overall outcome of MM patients. In vitro experiments do not fully replicate the complexity of the pathological mechanisms evident in animal models and/or in humans, but the results of our study justify further in vivo preclinical and clinical studies on the effects of carfilzomib-metformin coadministration on the cardiovascular system in the presence of comorbidities. Moreover, additional in vitro experiments aiming to elucidate molecular signaling events by experimental interfering are also needed, to fully elucidate the effects of carfilzomib and metformin on the vascular musculature.

## 4. Methods

### 4.1. Animals

Male C57BL/6J mice (12–14 of age) mice were bred and housed in the Translational Animal Research Center of the Johannes Gutenberg University, Mainz, Germany. All animal experiments were carried out in accordance with the “Guide for the care and use of Laboratory animals”, approved by the “Landesuntersuchungsamt Rheinland-Pfalz” and Ethics Committee of the University Medical Center of Johannes Gutenberg University, Mainz (G17-1-031, Germany, Mainz). Animals were purchased by Janvier Labs (Saint Berthevin Cedexand, France) and housed and maintained in specific pathogen free (SPF) cages (10 per cage; 25 ± 1 °C) at least for one week before the experiments, according to the Animal Research: Reporting of In Vivo Experiments (ARRIVE) guidelines [54,55]. Subsequently, mice were randomized as follows: Acute protocol: (i) control (NaCl 0.9%), (ii) Cfz (8 mg/kg), (iii) Met (140 mg/kg) and (iv) Cfz + Met (8 mg/kg, 140 mg/kg respectively) for 2 days [11], *n* = 5 per group. Subacute protocol: (i) control (NaCl 0.9%), (ii) Cfz (8 mg/kg), (iii) Met (140 mg/kg) and (iv) Cfz + Met (8 mg/kg, 140 mg/kg respectively) for 6 days [11], *n* = 10 per group. NaCl and Cfz were injected intraperitoneally on alternate days in the subacute protocol, while metformin was administered by oral gavage, daily, for 2 and 6 days, during the acute and subacute protocols respectively. Drug doses, treatment-duration and route of administration were established in our previous study addressing the cardiotoxicity of carfilzomib; therefore, is already established that by both a functional and a mechanistic approach the 4-dose carfilzomib administration regimen can appropriately recapitulate the clinical events manifested in treated patients [11]. At the end of the experiments, animals were anaesthetized by ketamine (100 mg/kg) and were euthanized by cervical dislocation. Blood samples and aortic tissue were collected. Blood samples were used for the determination of the leucocyte-derived reactive oxygen species (ROS) production, and aortic sections were subjected to ex vivo vascular studies and molecular analyses. For histology, cryosectioning and immunoblotting analyses, the thoracic part of the aortas was used by serial sectioning of aortic rings.

For the aged murine model 20 additional mice (15–17 months of age) were used for molecular analysis of the aortic tissue and serum NO_x_ content determination. The same procedures were used as in the young mice population in compliance with ARRIVE guidelines [54,55].

### 4.2. Vascular Relaxation and Constriction Studies

To assess the vasodilatory and vasoconstrictory potential of the murine aortas ex vivo, isolated aortic segments (3 mm) were mounted to force transducers in organ chambers to test their response to acetylcholine (ACh) and nitroglycerine (Gtn), used as vasodilatory agents in our settings. The aortic rings were preconstricted with prostaglandin F2a (3 nM) to reach 50–80% of the tone induced by KCl. Concentration-relaxation curves were recorded in response to the endothelium-dependent vasodilator ACh (1 nM to 3 mM) and smooth muscle dependent vasodilator Gtn (1 nM to 30 mM) as previously described [56].

### 4.3. Leucocyte Derived Oxidative Burst (ROS Release) Measurement

To stimulate the oxidative burst of leukocytes, whole blood was incubated for 85 min with phorbol 12,13-dibutyrate (PdBu) and Zymosan A as previously described [57]. PdBu, an activator of protein kinase C, leads to an unspecific oxidative burst from the blood leucocytes, while Zymosan A leads to a macrophage-derived ROS release. LO-12 a luminol-based chemiluminescent probe was used for the detection of ROS [57].

### 4.4. NO Metabolites (NO_x_) and 3-Nitrotyrosine Measurements

For serum nitrates-nitrites (NO_x_) measurements, sera were concentrated using 10 kDa cut-off filters and flow-through sample was further analyzed accordingly to the manufacturer’s instructions using a commercially available kit (No. 780001, Cayman Chemicals) [12].

For aortic 3-nitrotyrosine (3-NT) assessment, the abdominal part of the aortas for the treated animals was snap-frozen, pulverized and extracted with ice cold PBS (0.01 M, pH = 7.4). Subsequently samples were centrifuged at 5000× *g* for 15 min and lysate was further analyzed using a commercially available ELISA kit, accordingly the manufacturer’s instructions (3-nitrotyrosine ELISA Kit: No. K4158-100; BioVision) [12].

### 4.5. Histology

Aortic sections (5 mm) were fixed in 4% zinc-formalin for 24 h and sliced in 5 μm sections. Slides were deparaffinized in xylene and rehydrated in serial ethanol concentrations. Sections were hematoxylin–eosin and Sirius red stained for histology, as well as for wall thickness and collagen deposition assessment [57]. For immunofluorescence (IF) experiments aortic sections were snap-frozen in the optimal cutting temperature (OCT) medium (Tissue-Tek^®^ O.C.T.™ Compound, Fisher Scientific) and sliced in 3 μm cryosections [48].

### 4.6. Western Blot Immunoblotting Analysis

Thoracic aortic tissue was pulverized in a mortar in liquid nitrogen on dry ice and aortic powder was extracted with lysis buffer (1% Triton X-100, 20 mM Tris pH 7.4–7.6, 150 mM NaCl, 50 mM NaF, 1 mM EDTA, 1 mM EGTA, 1 mM glycerophosphatase, 1% SDS, 100 mM PMSF and 0.1% phosphatase-protease inhibitors cocktail). After centrifugation (11,000× *g*, 15 min, 4 °C) the supernatants were used to determine protein content, using a Lowry protein assay (Bio-Rad, Protein DC assay). After electrophoresis on a gradient SDS-PAGE gel (6–15%), proteins were transferred on polyvinylidene fluoride (PVDF) membranes (pore size 0.2 μm). Subsequently, the following primary antibodies were used: p-mTOR (mammalian target of rapamycin; Ser2448; mAb #5536), t-mTOR (mAb #2983), p-ACC (acetyl-CoA carboxylase; Ser79; mAb #11818), t-ACC (mAb #3676), p-Raptor (Ser792; mAb #2083), t-Raptor (mAb #2280), p-eNOS (endothelial nitric oxide synthase; Ser1177; mAb #9570), t-eNOS (mAb #32027), IRE1α (inositol-requiring enzyme 1 α; mAb #3294), iNOS (inducible nitric oxide synthase; mAb #13120), α-actinin (mAb #6487), Calnexin (mAb #2679), Bip (binding immunoglobulin protein; mAb #3177), p-AMPKα (AMP-activated kinase α subunit; Thr172; mAb #50081), t-AMPKα (mAb #5831), p-Akt (protein kinase B, Ser473; mAb #4060), t-Akt (pAb #9272), p-Beclin (Ser93; mAb #14717), t-Beclin (mAb #3495), p-Src (proto-oncogene tyrosine-protein kinase; Tyr416; mAb #6943), p-VASP (vasodilator-stimulated phosphoprotein; Ser239; pAb #3114), t-VASP (mAb #3132), α-Smooth Muscle Actin (αSMA; mAb #19245) and LC3B (microtubule-associated proteins 1A/1B light chain 3B; mAb #3868; Cell Signaling Technology, Beverly, MA, USA). PVDF membranes were incubated with secondary antibodies for 2 h at 24 °C (goat anti-rabbit HRP (# 7074); Cell Signaling Technology, Beverly, MA, USA) and developed using GE Healthcare ECL’s Western Blotting ECL (Thermo Scientific Technologies) chemiluminescent reagents. Relative densitometry was performed with appropriate software (NIH, USA) and the ratios were used for statistical analysis [11].

### 4.7. Analysis of Endothelial Markers via Real-Time PCR

For RNA isolation, snap-frozen thoracic murine aortas from the treated animals were pulverized and extracted by the standardized Trizol protocol. RT-PCR was performed with the CFX96 Real-Time PCR Detection System (Bio-Rad, Munich, Germany). Isolated RNA was reverse-transcribed to cDNA using high-capacity cDNA reverse transcription kit (Takara^®^). The following primer pairs were designed (Primer-Blast, NCBI, NIH) and used in order to detect mRNA expression of *icam-1, vcam-1, sele, tf, gadph, nox2* and *nox4* (Eurofins Genomics AT, GmbH) were analyzed using the SYBR^®^Green method (Kapa Biosystems) according to the manufacturer’s instructions [58]. Real-time PCR primers’ sequences are presented in Table 1**.**

### 4.8. Primary Murine Vascular Smooth Muscle Cell (PrmVSMCs) Isolation and Human Aortic Smooth Muscle Cell Culture

Mouse aortas were aseptically prepared and adipose tissue was removed. Aortas were cut in 1 mm rings and digested for 5-6 h in Dulbecco’s modified Eagle’s medium (DMEM, 10% fetal bovine serum (FBS)) supplemented with collagenase type II (1.42 mg/mL, # S8N10850, Worthington). Digestion mixture was plated on a 6-well plate for 5 days until VSMCs were confluent and then cells were subsequently split according to the experiment needs [59]. Primary human aortic smooth muscle cells (HAoSMCs, PromoCell) were seeded in 75mL flasks in optimal Smooth muscle cell growth medium (smooth muscle cell growth medium kit 2, PromoCell) until they reached confluency. In order to generate an in vitro model of senescence in the HAoSMCs, cells were sequentially split until passage 7 and were subsequently used for experimentation [15].

Cell viability and confluency assessment in the presence and absence of cardiovascular risk stimuli.

PrmVSMCs and HAoSMCs were seeded at a cell density of 6 × 10^3^ cells/well in 96-well plates. Cells were allowed to adhere and were incubated with Cfz (0.1, 0.3 µM) [14], Met (10 µM, 10 mM) [15] and the combination of the two drugs. Moreover, HAoSMCs were incubated at the optimum concentrations of the compounds, 24 h after preincubation with angiotensin-II (AngII, 100 nM) [16], with CoCl_2_ (150 µM) [17] and 48 h after preincubation with glucose (25 µM) [18] to simulate hypertensive, hypoxic and diabetic stimuli respectively. After the treatment cells were incubated in a solution of 3- (4,5-dimethylthiazol-2-yl) -2,5-biphenyl-tetrazole bromide (MTT; # M-5655, Sigma Aldrich, St. Louis, MO, USA) at a final concentration of 0.5 mg in DMEM for 4 h at 37 °C. Absorption was measured on a microplate spectrophotometer (Tecan Spark, Tecan Inc., Männedorf, Switzerland) at 570 nm (reference wavelength 690 nm) [60]. Confluency rate was measured by an automated brightfield cell imaging software of the microplate reader (Tecan Spark, Tecan Inc.).

### 4.9. In Vitro Study of the Cellular Metabolism—Oxygen Consumption Rate (OCR), Proton Efflux Rate (PER) and ATP Rate Measurements

Oxygen consumption rate (OCR) and proton efflux rate (PER) were measured in prmVSMCs and HAoSMCs, using an extracellular XF96 analyzer (Seahorse Bioscience, Agilent, CA, USA). The cells were seeded in 96-well XF 96-well cell culture plate (Seahorse Bioscience) at a density of 20 × 10^3^ cells/well in 200 µL DMEM and left for 5 h to attach to the plate. Subsequently, the cells were incubated with Cfz (0.1, 0.3 µM) [15] and the combination of the two drugs for 24 h. Growth medium was replaced by 180 µL/well preheated assay media (XF RPMI, Agilent) supplemented with 1 mM pyruvate, 2 mM L-glutamine and 4.5 mM or 7.4 mM D-glucose to simulate fasting or postprandial glucose levels, respectively. One hour after adaptation baseline OCR and PER was measured followed by measurements after the subsequent injection of oligomycin (Oligo, 1.5 mM) and of Antimycin-A and rotenone (Anti-A/Rot; each 0.5 μM) [61].

### 4.10. Immunofluorescence Confocal Microscopy

Cryosections of the aortas (5 µm) were fixed with 4% PFA and permeabilized by 0.1–0.2% Triton X-100 for 10 min. After blocking with 5% BSA, murine aorta was costained with α-Smooth Muscle Actin (αSMA; ab7817, Abcam, Germany) and LC3B (mAb #3868; Cell Signaling Technology, Beverly, MA, USA) antibodies. For vascular smooth muscle cells (VSMCs; prmVSMCs and HAoSMCs), unspecific binging of antibodies was blocked with 1% BSA followed by primary antibody incubation with the aforementioned primary monoclonal antibodies or with p53 (mAb #2527; Cell Signaling Technology, Beverly, MA, USA). After overnight incubation, sections were counterstained with the secondary antibodies (goat anti-mouse, ab150116; goat anti-rabbit IgG, ab150080, Abcam, Germany), stained with phalloidin-Alexa Fluor 488 (A12379, ThermoFisher Scientific Inc. Berkshire, UK) for 30 min and mounted in anti-fading mounting medium containing 4′,6-diamidino-2-phenylindole (DAPI; P36962, ThermoFisher Scientific Inc. Berkshire, UK) for confocal laser scanning. At least three individual images were acquired and fluorescence intensity of each probe was quantified by using FIJI/image J. For illustration purpose images were reconstructed in to 3D by using Imaris software [48].

### 4.11. Automated Immunofluorescence Microscopy

PrmVSMCs and HAOSMCs were imaged in phenol red free culture medium with an automated fluorescence microplate imager with a high-speed laser autofocus and environmental control to minimize phototoxicity due to manual focusing and to obtain unbiased brightness measurements. Imaging was conducted with an ArrayScanVTI imaging platform (ThermoFisher Scientific Inc. Berkshire, UK) as previously described [62]. For each probe, fluorescence of individual VSMC was quantified after background subtraction to account for uneven illumination by particle analysis using FIJI/ImageJ (LC3B (Cell Signaling Technology, Beverly, MA, USA)—goat anti-rabbit (ab150080, Abcam, Germany): Ex. 590 nm Em. 617 nm), α-tubulin (#62204, ThermoFisher Scientific Inc. Berkshire, UK)—goat anti-mouse (ab150116, Alexa-Fluor: Ex. 650 nm, Em. 665 nm, Abcam, Germany). To exclude debris, only particles larger than 200 pixels (low power field) were counted [63]. The cellular masks used for automated microscopy are presented in Appendix A.

### 4.12. Statistics

Statistical analysis was performed with GraphPad Prism 8 software and results were plotted as mean ± SEM values. Data were analyzed using a one-way ANOVA (Tukey post-hoc test for multiple comparisons) or two-way ANOVA (Tukey post-hoc test for multiple comparisons) as required. *p* values of *p* < 0.05 were considered statistically significant (* *p* < 0.05, ** *p* < 0.01, *** *p* < 0.001).

## Figures and Tables

**Figure 1 ijms-21-05185-f001:**
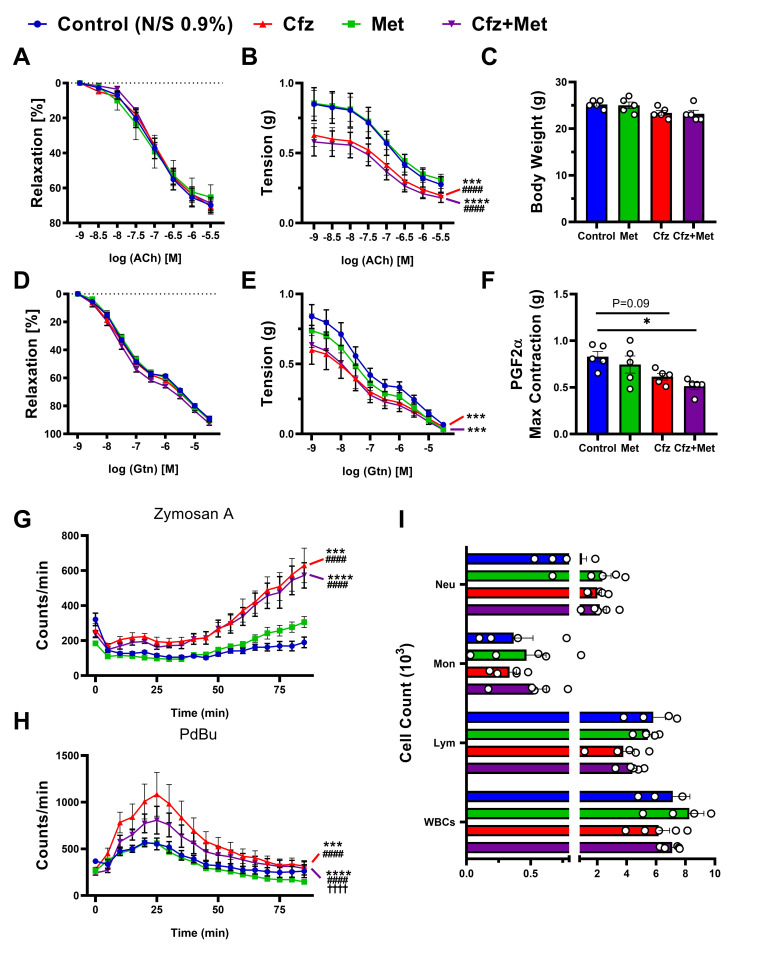
Carfilzomib leads to an acute hypo-contractile phenotype in murine vessels and a significant increase in leucocyte-derived reactive oxygen species (ROS) release, independently of metformin coadministration. Relaxation curves of (**A**) % relaxation to acetylcholine (ACh; *n* = 5 per group), (**B**) tension (g) following ACh-relaxation (*** *p* < 0.005 vs. control; #### *p* < 0.001 vs. Met; two way ANOVA, Tukey’s multiple comparison test, main column effect; *n* = 5 per group) and (**C**) graph of body weight (g; *n* = 5 per group). Relaxation curves of (**D**) % relaxation to nitroglycerine (Gtn) (n = 5 per group), (**E**) tension (g) following Gtn-relaxation (*** *p* < 0.005 vs. control; two way ANOVA, Tukey’s multiple comparison test, main column effect; *n* = 5 per group) and (**F**) PGF2α maximal contraction (g; * *p* < 0.05, one way ANOVA, Tukey’s multiple comparison test; *n* = 5 per group). Curves of (**G**) Zymosan A and (**H**) PdBu oxidative burst (counts/min) in course of time (min; *** *p* < 0.005 vs. control; #### *p* < 0.001 vs. Met; †††† *p* < 0.001 vs. Cfz; two way ANOVA, Tukey’s multiple comparison test, main column effect; *n* = 5 per group). (**I**) Graphs of circulating cell count (×10^3^) in whole blood samples. Data are presented as mean ± SEM and individual values are presented as scatter column graphs. Blue lines/columns represent the control (Normal Saline; NS 0.9%), red lines/columns represent Cfz (8 mg/kg, ip), green lines/columns Met (140 mg/kg, per os) and purple lines/columns Cfz+Met (8 mg/kg, ip and 140 mg/kg, per os, respectively) in the 2 days protocol (acute protocol). Cfz, carfilzomib; Met, metformin; Neu, Neutrophils; Mon, Monocytes; Lym, Lymphocytes; WBCs, White Blood Cells.

**Figure 2 ijms-21-05185-f002:**
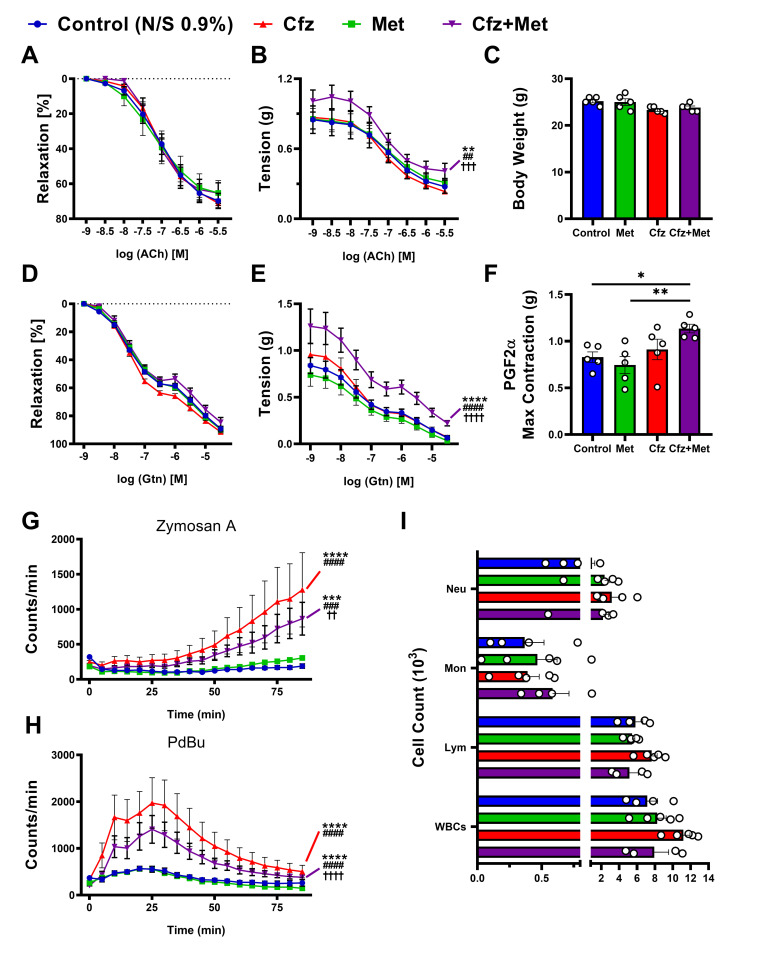
Carfilzomib did not exhibit any vascular deficits in subacute protocol, whilst it induced increased leucocyte derived-ROS release, which was partially inhibited by metformin. Coadministration of Cfz and Met led to an increased vascular reactivity to PGF2α. Relaxation curves of (**A**) % relaxation to acetylcholine (ACh; *n* = 5 per group), (**B**) tension (g) following ACh-relaxation (** *p* < 0.01 vs. control; ## *p* <0.01 vs. Met; ††† *p* < 0.005 vs. Cfz; two way ANOVA, Tukey’s multiple comparison test, main column effect; *n* = 5 per group) and (**C**) graph of body weight (g; *n* = 5 per group). Relaxation curves of (**D**) % relaxation to nitroglycerine (Gtn; *n* = 5 per group), (**E**) tension (g) following Gtn-relaxation (**** *p* < 0.001 vs. control; #### *p* <0.001 vs. Met; †††† *p* < 0.001 vs. Cfz two way ANOVA, Tukey’s multiple comparison test, main column effect; *n* = 5 per group) and (**F**) PGF2α maximal contraction (g) (* *p* < 0.05, ** *p* < 0.01; one way ANOVA, Tukey’s multiple comparison test; *n* = 5 per group). Curves of (**G**) Zymosan A and (**H**) PdBu oxidative burst (counts/min) in course of time (min; *** *p <* 0.005 and **** *p* < 0.001 vs. control; *### p<* 0.005 and #### *p* < 0.001 vs. Met; †† *p* < 0.01, †††† *p* < 0.001 vs. Cfz; two way ANOVA, Tukey’s multiple comparison test, main column effect; *n* = 5 per group). (**I**) Representative graphs of circulating cell count (×10^3^) in whole blood samples. Data are presented as mean ± SEM and individual values are presented as scatter column graphs. Blue lines/columns represent control (Normal Saline; NS 0.9%), red lines/columns Cfz (8 mg/kg, ip), green lines/columns Met (140 mg/kg, per os) and purple lines/columns Cfz+Met (8 mg/kg, ip and 140 mg/kg, per os, respectively) in the 6 days protocol (subacute protocol). Cfz, Carfilzomib; Met, Metformin; Neu, Neutrophils; Mon, Monocytes; Lym, Lymphocytes; WBCs, White Blood Cells.

**Figure 3 ijms-21-05185-f003:**
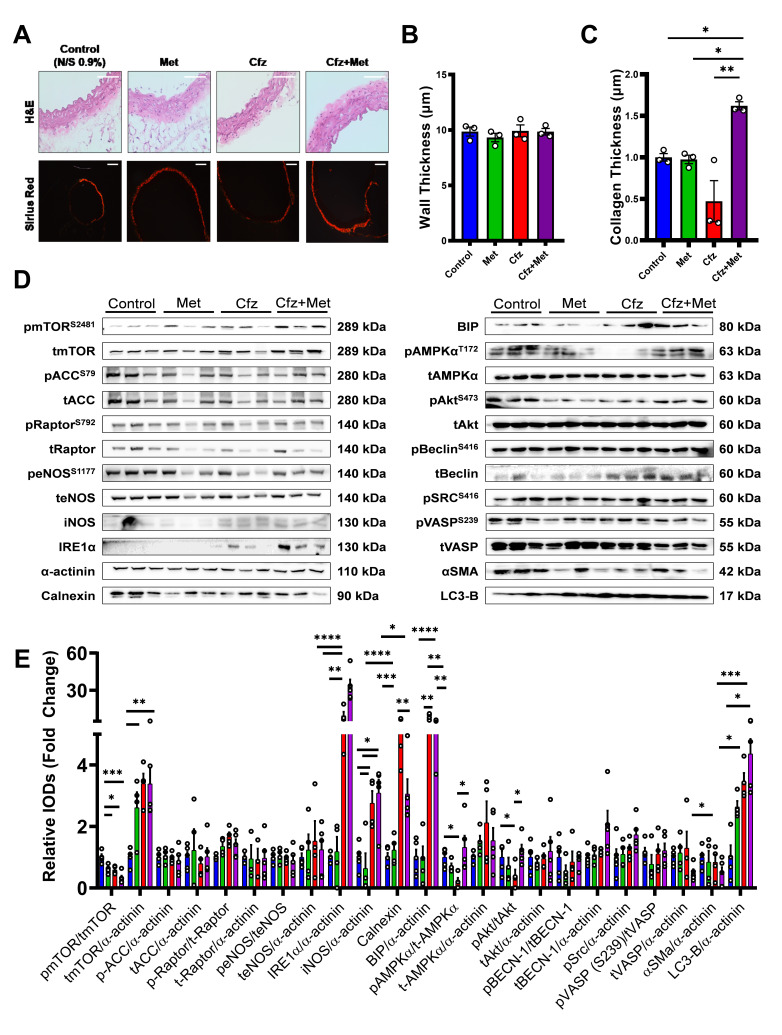
Carfilzomib and metformin monotherapies do not induce any changes in the vascular phenotype, while the combination of Cfz+Met leads to increased collagen deposition on the vessels and a synergistic induction of an ER-stress/AMPKα/LC-3B dependent autophagy. (**A**). (Upper Panel) Representative images of hematoxylin–eosin (H&E) staining of murine aortas (magnification 40×; scale bar 40 μm; Lower Panel) Representative images of Sirius red staining (magnification 20×; scale bar 40 μm). Graphs of (**B**) aortic wall thickness (μm) and (**C**) collagen thickness (μm) (* *p* < 0.05, ** *p* < 0.01, one way ANOVA, Tukey’s multiple comparison test; *n* = 3 per group). (**D**) Representative Western blot images of the studied protein targets and their phosphorylated forms. (**E**) Relative densitometry analysis of Western blot analysis, presented as fold change of the control (* *p* < 0.05, ** *p* < 0.01, *** *p* < 0.005, **** *p* < 0.001; one way ANOVA, Tukey’s multiple comparison test; *n* = 3 per group). Data are presented as mean ± SEM and individual values are presented as scatter column graphs. Blue lines/columns represent the control (Normal Saline; NS 0.9%), red lines/columns represent Cfz (8 mg/kg, ip), green lines/columns Met (140 mg/kg, per os) and purple lines/columns Cfz+Met (8 mg/kg, ip and 140 mg/kg, per os, respectively) in the 6 days protocol (subacute protocol). Cfz, Carfilzomib; Met, Metformin.

**Figure 4 ijms-21-05185-f004:**
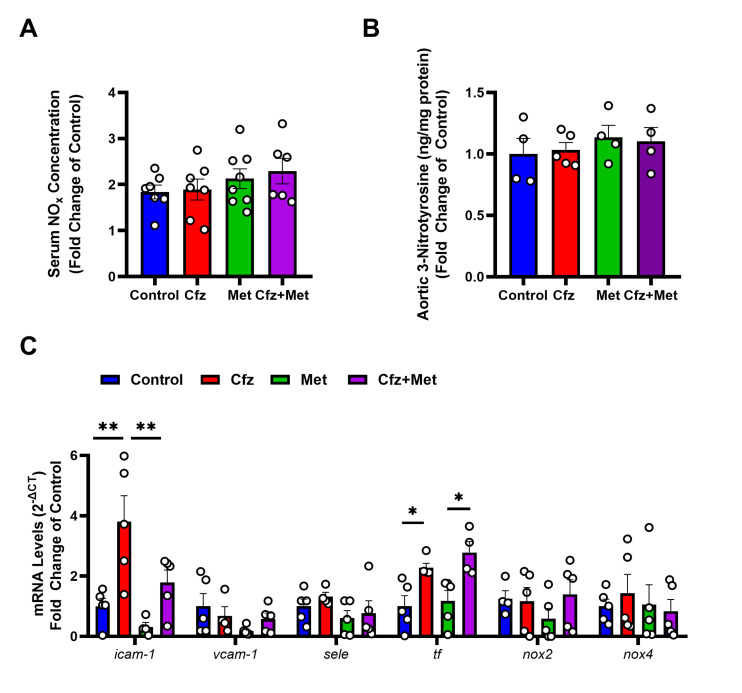
Carfilzomib did not affect the NO pathway but increased icam-1 and tf expression. Metformin partially abrogated icam-1 mRNA increase. Graphs of (**A**) serum NOx concentration (μM; *n* = 5–7 per group) and (**B**) graph of aortic 3-nitrotyrosine (ng/mg protein; fold change of the control) of treated mice (*n* = 5 per group). (**C**) RT-PCR mRNA expression of proinflammatory and regulatory endothelial genes and redox molecules (*n* = 4–5 per group; * *p* <0.05, ** *p* < 0.01; one way ANOVA, Tukey’s multiple comparison test). Data are presented as mean ± SEM and individual values are presented as scatter column graphs. Blue lines/columns represent the control (Normal Saline; NS 0.9%), red lines/columns represent Cfz (8 mg/kg, ip), green lines/columns Met (140 mg/kg, per os) and purple lines/columns Cfz+Met (8 mg/kg, ip and 140 mg/kg, per os, respectively) in the 6 days protocol (subacute protocol). Cfz, Carfilzomib; Met, Metformin.

**Figure 5 ijms-21-05185-f005:**
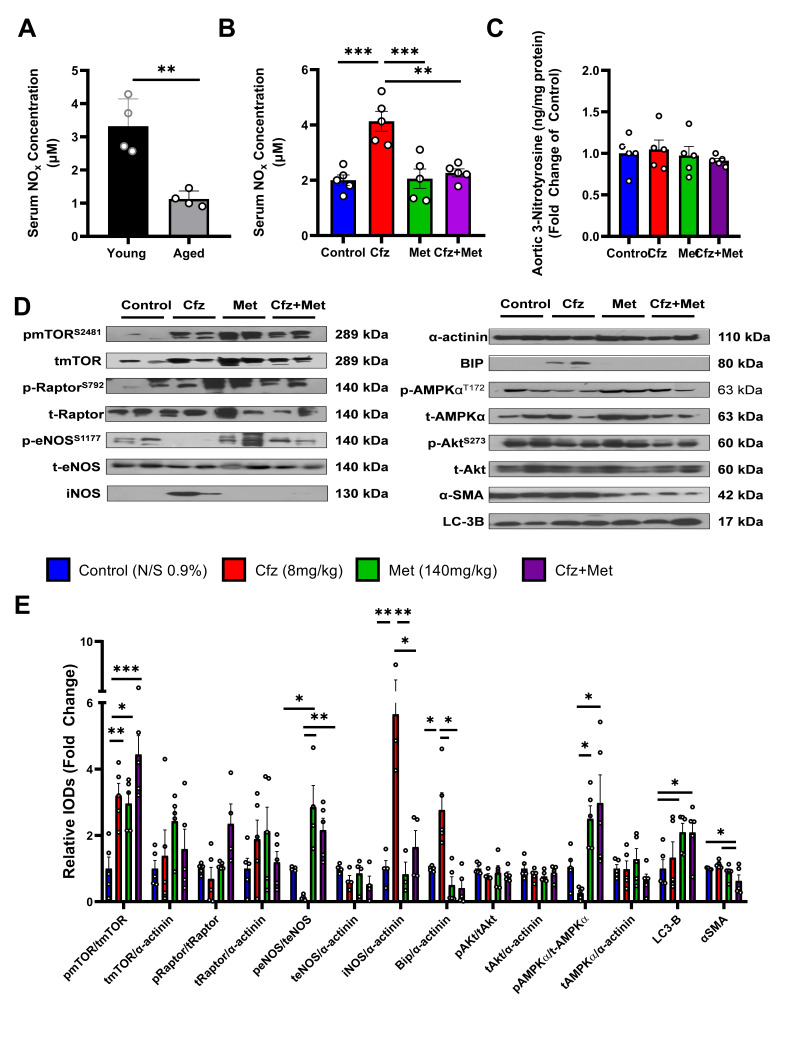
Carfilzomib and metformin coadministration induced LC3-B-dependent autophagy in the aortas of aged mice, while Cfz increased serum NO_x_ content without leading to a nitro-oxidative stress in the aortas. Graphs of serum NO_x_ concentration (μM) of (**A**) young (3–4 months of age) and aged (15–17 months of age) mice (*n* = 4 per group). (**B**) Aged mice treated with carfilzomib (Cfz), metformin (Met) or the combination of the drugs (*n* = 5 per group). (**C**) Graph of aortic 3-nitrotyrosine (3-NT; ng/mg protein; fold change of the control) of aged treated mice (*n* = 5 per group). (**D**) Representative Western blot images of aged murine aortas and (**E**) relative densitometry analysis of the selected proteins and their phosphorylated forms (*n* = 5 per group; * *p* < 0.05, ** *p* < 0.01, *** *p* < 0.005, one way ANOVA, Tukey’s multiple comparison test). Data are presented as mean ± SEM and individual values are presented as scatter column graphs. Blue lines/columns represent the control (Normal Saline; NS 0.9%), red lines/columns represent Cfz (8 mg/kg, ip), green lines/columns Met (140 mg/kg, per os) and purple lines/columns Cfz+Met (8 mg/kg, ip and 140 mg/kg, per os, respectively) in the 6 days protocol (subacute protocol). Cfz, Carfilzomib; Met, Metformin.

**Figure 6 ijms-21-05185-f006:**
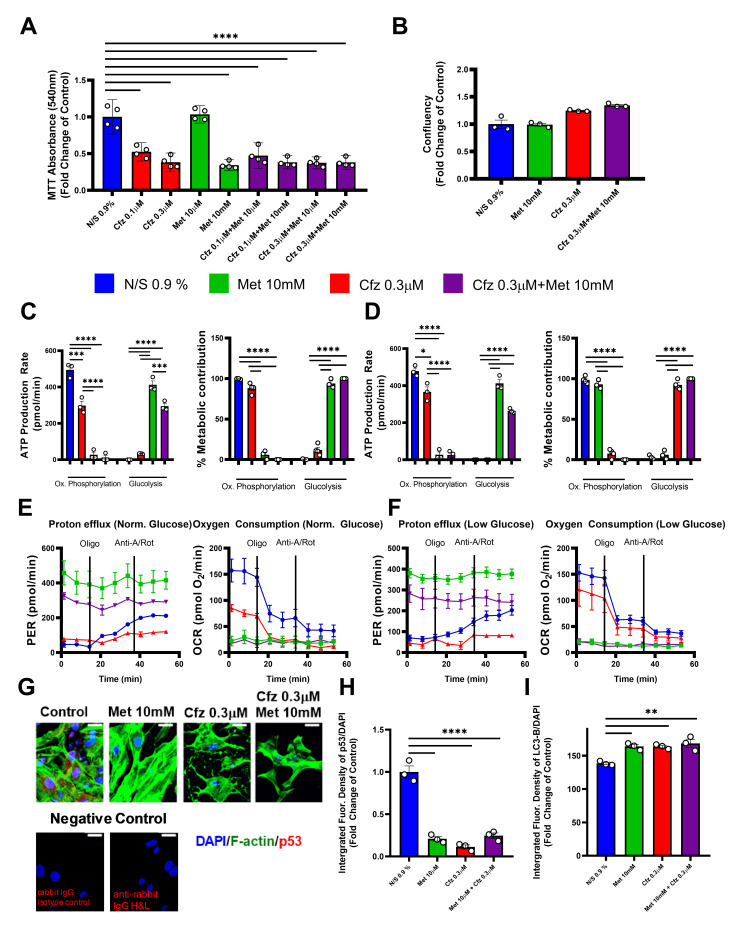
Carfilzomib and metformin induce cytotoxicity in primary murine vascular smooth muscle cells (prmVSMCs) in vitro. Graphs of (**A**) MTT absorbance (540 nm) expressed as the fold change of the control (**** *p* < 0.001; one way ANOVA, Tukey’s multiple comparison test; *n* = 4 per group), (**B**) cellular confluency expressed as the fold change of the control (*n* = 3 per group), (**C**) ATP production rate (pmol/min) and % metabolic contribution under conditions of normal glucose (10 mM) originating from oxidative phosphorylation and glycolysis (*** *p* < 0.005, **** *p* < 0.001; one way ANOVA, Tukey’s multiple comparison test; *n* = 3 per group) and (**D**) ATP production rate (pmol/min) and % metabolic contribution under conditions of low glucose (1 mM) originating from oxidative phosphorylation and glycolysis (* *p* < 0.05, **** *p* < 0.001; one way ANOVA, Tukey’s multiple comparison test; *n* = 3 per group). Time-course graphs of the proton efflux rate (PER) and oxygen consumption rate (OCR) under the conditions of (**E**) normal glucose (10 mM) and (**F**) low glucose (1 mM), as studied with ATPase inhibitor oligomycin (Oligo) and cytochrome c reductase inhibitor antimycin-a (Anti-A) and complex I inhibitor rotenone (Rot). (**G**) Merged immunofluorescence confocal images of prmVSMCs stained against F-actin (Phalloidin, green), p53 (red) and DAPI (blue) as well as the negative control for the staining (rabbit isotype control and anti-rabbit IgG H&L secondary antibody). Graphs of (**H**) integrated fluorescence density of p53/DAPI expressed as the fold change of Normal Saline (NS) 0.9%, serving as the control vehicle and (**I**) total fluorescence intensity of LC3B/DAPI as measured in the automated microscope per well (mean total fluorescence intensity of individual VSMCs in the well; ** *p* < 0.01, **** *p* < 0.001; one way ANOVA, Tukey’s multiple comparison test; *n* = 3 per group). Data are presented as mean ± SEM and individual values are presented as scatter column graphs. Blue lines/columns represent Normal Saline (NS) 0.9%, red lines/columns represent Cfz (0.3 μΜ), green lines/columns Met (10 mM) and purple lines/columns Cfz+Met (0.3 μΜ, 10 mM respectively). Cfz, carfilzomib; Met, metformin; prmVSMCs, primary murine vascular smooth muscle cells.

**Figure 7 ijms-21-05185-f007:**
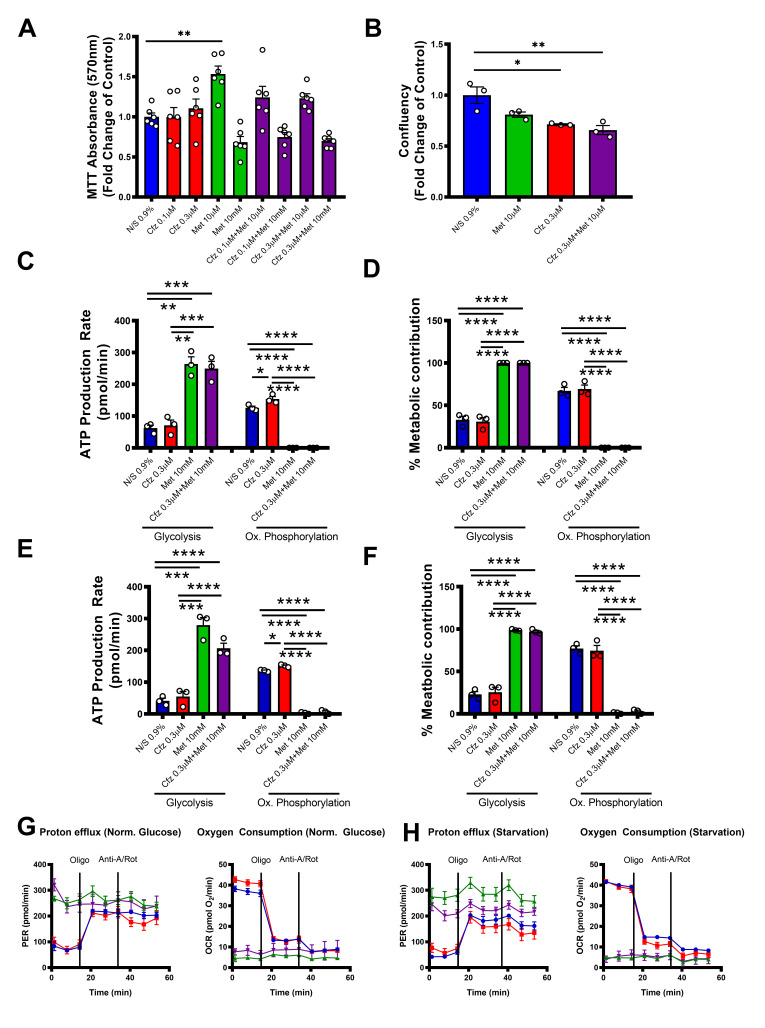
Carfilzomib and Cfz+Met do not induce cytotoxicity in senescent human aorta smooth muscle cells in vitro. Met shifts cellular metabolism to glycolysis. Graphs of (**A**) MTT absorbance (540 nm) expressed as the fold change of Normal Saline (NS) 0.9% (** *p* < 0.01; one way ANOVA, Tukey’s multiple comparison test; *n* = 6 per group), (**B**) cellular confluency expressed as the fold change of Normal Saline (NS) 0.9% (*n* = 3 per group), (**C**) ATP production rate (pmol/min) and (**D**) % metabolic contribution under conditions of normal glucose (10 mM) originating from oxidative phosphorylation and glycolysis, (**E**) ATP production rate (pmol/min) and (**F**) % metabolic contribution under conditions of low glucose (1 mM) originating from oxidative phosphorylation and glycolysis (* *p* < 0.05,** *p* < 0.01, *** *p* < 0.01, **** *p* < 0.001; one way ANOVA, Tukey’s multiple comparison test; *n* = 3 per group). Time-course graphs of the proton efflux rate (PER) and oxygen consumption rate (OCR) under conditions of (**G**) normal glucose (10 mM) and (**H**) low glucose (1 mM), as studied with ATPase inhibitor oligomycin and cytochrome c reductase inhibitor antimycin-a and complex I inhibitor rotenone. Data are presented as mean ± SEM and individual values are presented as scatter column graphs. Blue lines/columns represent Normal Saline (NS) 0.9%, red lines/columns represent Cfz (0.3 μΜ), green lines/columns Met (10 mM) and purple lines/columns Cfz+Met (0.3 μΜ, 10 mM respectively). Cfz, carfilzomib; Met, metformin; HAoSMCs, human aorta smooth muscle cells; Rot, rotenone; Oligo, oligomycin; Anti-A, antimycin A.

**Figure 8 ijms-21-05185-f008:**
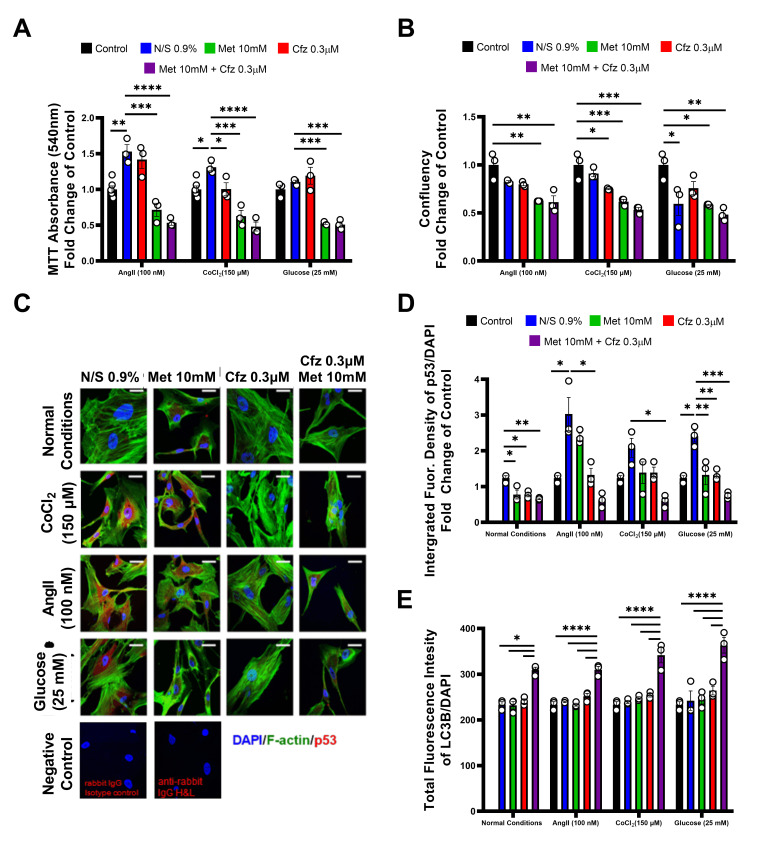
Cfz+Met synergistically induce autophagy in senescent human aorta smooth muscle cells (HAoSMCs) in the presence of cardiovascular risk stimuli in vitro. Graphs of (**A**) MTT absorbance (540 nm, *n* = 3–5 per group) and (**B**) cellular confluency expressed as the fold change of the control (untreated senescent HAoSMCs; * *p* < 0.05, ** *p* < 0.01, *** *p* < 0.01, **** *p* < 0.001; one way ANOVA, Tukey’s multiple comparison test; *n* = 3 per group). (**C**) Representative immunofluorescence confocal images of HAoSMCs under normal conditions (untreated cells) or in the presence of AngII (100 nM), CoCl2 (150 μΜ) and high-glucose (25 mM). (**D**) Integrated fluorescence density of p53/DAPI expressed as the fold change of the control (untreated senescent HAoSMCs) and (**E**) total fluorescence intensity of LC3B/DAPI as measured in the automated microscope per well (mean total fluorescence intensity of individual VSMCs in the well; * *p* < 0.05, ** *p* < 0.01, *** *p* < 0.005, **** *p* < 0.001; one way ANOVA, Tukey’s multiple comparison test; *n*=3 per group). Data are presented as mean ± SEM and individual values are presented as scatter column graphs. Black columns represent the control (untreated cells), blue columns represent Normal Saline (NS) 0.9%, red columns represent Cfz (0.3 μΜ), green columns Met (10 mM) and purple columns Cfz+Met (0.3 μΜ, 10 mM respectively). Cfz, carfilzomib; Met, metformin; HAoSMCs, human aorta smooth muscle cells.

**Figure 9 ijms-21-05185-f009:**
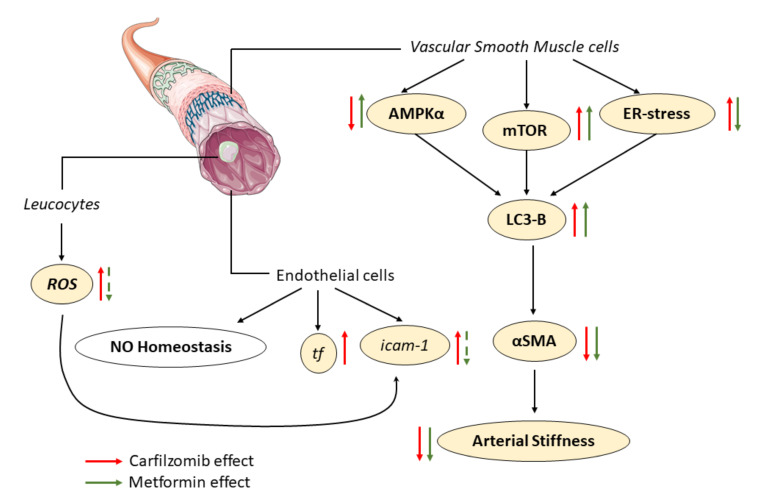
Schematic representation of the effect of carfilzomib and metformin coadministration on the vessels. White panels represent independent molecular targets, red arrows indicate Cfz-dependent pathways, green arrows represent Met-dependent pathways. Cfz, carfilzomib; Met, metformin.

**Table 1 ijms-21-05185-t001:** Real-Time PCR primers sequences.

Gene Name	Forward	Reverse	Product Length	Tm
*Mus musculus* intercellular adhesion molecule 1 (icam1)	AGTCCGCTGTGCTTTGAGAA	CTCTCCGGAAACGAATACACG	80	59.8/58.8
*Mus musculus* vascular cell adhesion molecule 1 (vcam1)	TCTTTATGTCAACGTTGCCCCC	ACTTGAGCAGGTCAGGTTCAC	97	61.1/60.2
*Mus musculus* E-selectin	TCTGCAGTTCTGACGTGTGG	AGTGCAACTACCAAGGGACG	98	60.2/60.0
*Mus musculus* coagulation factor III (F3)	CGTGAAGGATGTGACCTGGG	TCACAAGTTGGTCTCCGTCC	89	60.4/59.6
*Mus musculus* glyceraldehyde 3-phosphate dehydrogenase (gadph)	CCCAGCTTAGGTTCATCAGGT	GCCAAATCCGTTCACACCG	87	59.4/59.7
*Mus musculus* nadph oxidase 2 (NOx2)	AAGTTCGCTGGAAACCCTCC	GCCAAAACCGAACCAACCTC	88	60.3/60.0
*Mus musculus* nadph oxidase 2 (NOx4)	CACCAAATGTTGGGCGATTGT	CAGGACTGTCCGGCACATAG		60.0/60.2

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
