# Peer review of "Investigating the Vascular Toxicity Outcomes of the Irreversible Proteasome Inhibitor Carfilzomib"

_ijms, 2020, doi:10.3390/ijms21155185_

Round 1

Reviewer 1 Report

Efentakis et al. have investigated the effects of the proteasome inhibitor Carfilzomib on vessel physiology ex vivo and in vivo, and its effect on VSMC metabolism and autophagy in vitro. They also assessed the effect of a co-administration of the drug metformin, which has a protective effect on carfilzomib induced cardiotoxicity, to elucidate if a co-administration would be beneficial for patients treated with carfilzomib. The authors have performed extensive in vivo, ex vivo and in vitro analyses. The manuscript is well written and scientifically sound with a clear research question. The authors also discuss their findings very well importantly including an interpretation of the sometimes contradictive findings.

I have some points, which should be clarified to improve the manuscript:

In the discussion the authors state that carfilzomib treatment had no permanent vascular or endothelial damage. They also state that Cfz does not seem to cause endothelial dysfunction. I presume the authors base this statement on the findings of Figure 3D where no influence on pVASP, VASP, peNOS, eNOS could be seen. Endothelial damage or dysfunction is much more than activation of eNOS and VASP. In my opinion, to be able to make this statement, the authors need to assess for instance ROS production from the endothelium or the upregulation of inflammatory and pro-thrombotic markers, such as ICAM, VCAM, Selectins or TF, or endothelial cytotoxicity by MTT etc.

In the discussion the authors state that “Co-administration of the drugs led to the restoration of AMPKα phosphorylation and increased LC-3B-dependent autophagy in an AMPKα/ER-stress dependent pathway, as well as through metabolic changes in SMCs both in vivo and in vitro (Figure 7), indicating that the co-administration of the two drugs may exert vasoprotection.” I find this expression somewhat confusing. Co-administration of met and Cfz caused an increase in PMN induced ROS (even if Met could significantly reduce the ROS-production, it is still a significant increase versus control and almost as high as with Cfz alone, especially with Zymosan A stimulation), increased cytotoxicity of HAoSMC and higher collagen disposition in the vessels. I am not convinced that one can draw the conclusion that Met + Cfz treatment is vasoprotective. Furthermore, the authors state later in the discussion that “Carfilzomib and metformin exhibited cytotoxicity through metabolic alterations and autophagy in compliance with our in vivo experiments.” This statement stands in direct contrast to the previous one. Could the authors please clarify the situation.

Why did the authors measure ROS production of whole blood and not in the isolated vessels (in addition)? It would be very interesting to assess the effect of Cfz on ROS production in the endothelium and in VSMC.

Figure 3: It was nearly impossible to read the text in the figure (to the Western blot and in immunofluorescence images) due to bad resolution. This may be due to the PDF-conversion, but the text is nevertheless too small. The blot of mTOR phosphorylation is not very representative to the quantitative analysis. mTOR phosphorylation seems to rather increase with Cfz + Met treatment, but this is hard to interpret, as tmTOR bands varies (even within the same samples). If the authors have a more representative blot, I would recommend using it. The same goes for LCB-3, which is more like a smear. Please state in the figure legend what material was used for the blots (whole vessels/aortas).

Figure 4: Representative images for immunofluorescent staining of LC3-B are not shown, only for p53. Why? 4I: The y-axis of the graph states p53/DAPI. In the figure legend, 4I refers to LCB-3/DAPI. Please correct. The y-axis also contains a spelling mistake.

In the text to the results of figure 6, the authors state that “comorbidity stimuli led to an increase of p53 expression in HAoSMC”. Looking at the graph in figure 6B, I cannot see any change in the control sample between non stimulated cells and cells stimulated with AngII, CoCl2 or high glucose. Please explain.

Please write the full term/name (for instance for PGF2a), the first time it is used and not only the abbreviation.

Materials and method section:

Please add details to the ROS release measurement. What was used to measure ROS release (DCF, L-O12…..)?

Please add the source of the human aortic smooth muscle cells.

Author Response

We would like to thank the Editors and Reviewers for evaluating our manuscript. During the revision process, we have addressed all comments raised point by point which undoubtedly provided us with a valuable opportunity to improve our manuscript. We hope that the revised manuscript will be acceptable for publication and will be of interest to the readers of International Journal of Molecular Sciences.

We have marked the modified texts as highlighted yellow in the revised manuscript.

Reviewer 1

Efentakis et al. have investigated the effects of the proteasome inhibitor Carfilzomib on vessel physiology ex vivo and in vivo, and its effect on VSMC metabolism and autophagy in vitro. They also assessed the effect of a co-administration of the drug metformin, which has a protective effect on carfilzomib induced cardiotoxicity, to elucidate if a co-administration would be beneficial for patients treated with carfilzomib. The authors have performed extensive in vivo, ex vivo and in vitro analyses. The manuscript is well written and scientifically sound with a clear research question. The authors also discuss their findings very well importantly including an interpretation of the sometimes-contradictive findings.

Reply: We would like to thank reviewer for their kind comments, which were really helpful in improving our manuscript.

Comment 1

In the discussion the authors state that carfilzomib treatment had no permanent vascular or endothelial damage. They also state that Cfz does not seem to cause endothelial dysfunction. I presume the authors base this statement on the findings of Figure 3D where no influence on pVASP, VASP, peNOS, eNOS could be seen. Endothelial damage or dysfunction is much more than activation of eNOS and VASP. In my opinion, to be able to make this statement, the authors need to assess for instance ROS production from the endothelium or the upregulation of inflammatory and pro-thrombotic markers, such as ICAM, VCAM, Selectins or TF, or endothelial cytotoxicity by MTT etc.

Reply: We thank the reviewer for this important point. In order to support our data on the direct effect of Cfz on the endothelium, we performed new experiments on additional 20 C57Bl/6 young mice and 20 aged (15-17 months old) mice. Following reviewer’s suggestion, we performed RT-PCR analysis of endothelial pro-inflammatory genes, namely intercellular adhesion molecule-1 (icam-1) and vascular cell adhesion molecule 1 (vcam-1), endothelial specific selectin selectin e (sele) and tissue factor (tf). Moreover, in order to investigate the circulating NO homeostasis, we assessed nitrate and nitrite (NOx) concentration in the serum. We found that in young C57Bl/6 mice, carfilzomib did not alter NO metabolites in the serum in compliance with eNOS and VASP phosphorylation status. Endothelial regulatory genes such as sele and vcam-1 were found to be unchanged among groups. However, carfilzomib was found to upregulate icam-1 mRNA levels, an effect partially mitigated by metformin, and increased tf mRNA levels, independently of metformin administration. Therefore, it is possible that even though carfilzomib does not induce a dysregulation of NO homeostasis in the endothelium (therefore not affecting vasorelaxation and vasoconstriction), it might induce an endothelial-derived proinflammatory and pro-thrombotic phenotype, which is not directly connected with vascular deficits.

The aforementioned data are presented in the results section page14; line 214-222 and in Figure 4 of the revised manuscript as follows:

“Subsequently, we investigated the endothelial homeostasis of the treated murine aortas. Using Real-Time PCR (RT-PCR) analysis we assessed key pro-inflammatory and regulatory genes of the endothelial cells which included intercellular adhesion molecule 1 (icam1), vascular adhesion molecule 1 (vcam1), endothelial selectin (sele) and tissue factor (tf). Carfilzomib caused an increase in icam-1 mRNA levels in the aortas, partially abrogated by metformin administration and an upregulation of tf mRNA levels, which was independent of metformin administration. Vcam-1 and sele, did not differ among study groups. Therefore, carfilzomib might lead to a pro-inflammatory and pro-thrombotic phenotype of the vessels, as shown by icam-1 and tf respectively, which requires further investigation”.

and discussed in pages 30-31; lines 441-464 as follows:

“Moreover, carfilzomib did not alter NO homeostasis as assessed by circulating NOx levels nor led to nitro-oxidative damage or redox alterations in the aortas, as shown by the stable 3-NT and nox2-nox4 mRNA levels. On the contrary, carfilzomib increased icam-1 mRNA levels, revealing a pro-inflammatory phenotype of the vessels. It has been previously demonstrated that the increased circulating ROS, as observed in our protocol in the Cfz-treated leucocytes (Figure 2G-H), can induce endothelial ICAM-1 expression [35]. Therefore, circulatory oxidative stress can be accredited for icam-1 mRNA upregulation in Cfz-treated aortas. Histological evaluation of the aortas (Figure 3A) did not however provide any evidence for immune cell infiltration of the aortas. Therefore, the induction of inflammation in the sub-acute protocol cannot be established.  In agreement with the ROS-releasing capacity of the leucocytes isolated by mice treated with the combination of carfilzomib and metformin, the drug’s combination seems to partially abrogate icam-1, as a result of the mitigated oxidative stress in the circulation. This mild pro-inflammatory phenotype might be of interest in a chronic model of carfilzomib treatment, aimed at investigating the effect of the drug on vascular function in the long term.

Concerning the regulation of TF (tissue factor) of the aortas, we found that carfilzomib increased tf mRNA levels, independently of metformin treatment. TF is a key regulatory integral membrane protein on the surface of endothelial cells, implicated in atherosclerosis, and a major procoagulant for thrombus formation associated with endothelial dysregulation [36]. To the best of our knowledge, there are no data on the effect of carfilzomib on TF regulation. However, there are data supporting a prothrombotic activity of carfilzomib, as implicated by the common complication of acute thrombotic microangiopathy (TMA) seen in carfilzomib treated patients [37, 38]. It is, therefore, possible that carfilzomib induces a (pro-)thrombotic phenotype in the vessels, but investigating this action was not within the scope of the current study. The effect of carfilzomib on the coagulation phenotype and on microvascular thrombosis will be addressed in a future research work.”

Comment 2

In the discussion the authors state that “Co-administration of the drugs led to the restoration of AMPKα phosphorylation and increased LC-3B-dependent autophagy in an AMPKα/ER-stress dependent pathway, as well as through metabolic changes in SMCs both in vivo and in vitro (Figure 7), indicating that the co-administration of the two drugs may exert vasoprotection.” I find this expression somewhat confusing. Co-administration of met and Cfz caused an increase in PMN induced ROS (even if Met could significantly reduce the ROS-production, it is still a significant increase versus control and almost as high as with Cfz alone, especially with Zymosan A stimulation), increased cytotoxicity of HAoSMC and higher collagen disposition in the vessels. I am not convinced that one can draw the conclusion that Met + Cfz treatment is vasoprotective. Furthermore, the authors state later in the discussion that “Carfilzomib and metformin exhibited cytotoxicity through metabolic alterations and autophagy in compliance with our in vivo experiments.” This statement stands in direct contrast to the previous one. Could the authors please clarify the situation.

Reply: We thank the reviewer for this comment and we agree with the reviewer that the provided data were not sufficient to make the afore-mentioned statement. Cytotoxicity in the smooth muscle cells is vasoprotective, only in terms of cardiovascular comorbidities or risk-factor-derived hypertrophy of the vessels. To resolve this concern we have added additional mechanistic experiments on aged murine aortas investigating the molecular mechanism of carfilzomib and metformin. In these experiments and in line with what we have already shown in vivo and in vitro, the co-administration of the drugs leads to a synergistic induction of LC3-B-dependent autophagy and a decrease in α-smooth muscle actin (αSMA).

Please find the new data on the revised manuscript in the results section in page 16; lines 236-255 and in Figure 5 as follows:

“Carfilzomib increases NOx content in the serum of the aged mice and upregulates iNOS and BIP in the aortic tissue. Co-administration of carfilzomib and metformin leads to an induction of LC-3B-dependent autophagy in the aged murine aortas 

Given the higher incidence of MM among the elderly population [5], we also assessed the effects of sub-acute carfilzomib and metformin treatment in an aged murine model. As per the data in young mice, carfilzomib did not induce nitro-oxidative damage in the aortas, as demonstrated by the unchanged levels of 3-NT (Figure 5C). However, aged mice exhibited decreased serum NOx levels, and carfilzomib administration increased circulating NOx (Figure 5A-B) which was associated with an upregulation of iNOS in the aortic tissue. In agreement with the mechanistic data from the young murine aortas, carfilzomib upregulated BIP, a marker of unfolded protein response. Carfilzomib and the co-administration of carfilzomib and metformin did not affect Raptor phosphorylation or expression but led to an increase in mTOR inhibitory phosphorylation at S2481. Metformin increased eNOS phosphorylation compared to the control group, and metformin as well as the drug co-administration increased both eNOS and AMPKα phosphorylation compared to carfilzomib alone. Furthermore, metformin and Cfz+Met led to increased LC3-B expression (Figure 5D-E). Importantly and in compliance with the young mice data, the drugs’ co-administration led to a decrease in αSMA.

Collectively, the mechanistic data obtained from both the young and the aged murine models show that sub-acute co-administration of carfilzomib and metformin induces a LC3-B-dependent autophagy in the vessels which is independent of age. These effects are observed mostly in the vascular smooth muscle cell layer of the aortas.” 

and in the discussion section in pages 31-32; lines 488-502 as follows:

“To increase the translational value of our experiments, we incorporated an aged murine model to investigate the mechanistic effects of carfilzomib on the aged vasculature. In agreement with data from the young mice, carfilzomib did not induce nitro-oxidative damage on the vessels as demonstrated by its surrogate marker 3-NT, despite the increased serum NOx concentration and iNOS expression seen in the carfilzomib-treated aged aortas. Mechanistically, metformin increased AMPKα and eNOS phosphorylation -independently of carfilzomib treatment- an effect that is already shown to be vasoprotective in aged vessels [45]. Carfilzomib and metformin co-administration led to increased AMPKα and eNOS phosphorylation, as well as to a synergistic induction of autophagy as shown by mTOR inhibitory phosphorylation at S2481, and increased LC3-B expression. Moreover, a significant decrease of αSMA – a surrogate marker of the VSMCs- was observed only in the Cfz+Met group. The induction of autophagy in aged vessels ameliorates vascular hypertrophy and exerts vasoprotective effects, by decreasing αSMA expression as a result of decreased VSMCs proliferation [46]. Therefore, carfilzomib and metformin co-administration exerts a significant vasoprotective effect on the aged vessels via increased vascular LC3-B-dependent autophagy and increased eNOS and AMPKα phosphorylation.”

Moreover, we have rephrased the discussion section, please see page 28; lines 409-412 in the revised manuscript, as follows: “These findings indicate that the effect of carfilzomib on the vasculature is mainly vascular smooth muscle cell-dependent and involves the regulation of LC3B-autophagy, which is further amplified by metformin”.    

Comment 3

Why did the authors measure ROS production of whole blood and not in the isolated vessels (in addition)? It would be very interesting to assess the effect of Cfz on ROS production in the endothelium and in VSMC.

Reply: We thank the reviewer for this comment. In order to assess the effect of Cfz on nitro-oxidative damage in the aortas, we performed additional experiments, in both young and aged mice, measuring 3-nitrotyrosine (3-NT) concentration. We found that both in the young and aged mice, carfilzomib did not induce aortic nitro-oxidative damage. Moreover, through RT-PCR analysis we assessed the mRNA levels of key redox regulating enzymes, namely nox2 and nox4, which are known to mediate ROS release and induce oxidative stress. We found that carfilzomib did not increase neither nox2 nor nox4 mRNA levels in the aortic tissue, revealing that carfilzomib does not interfere with the redox regulation of the aortas.

Please refer to page 14; lines 202-213 in the results section and in Figure 4 as follows:

Carfilzomib treatment did not affect endothelial homeostasis and did not induce nitro-oxidative stress in the murine aortas

To further investigate the endothelial homeostasis as well as the redox status of the carfilzomib- and metformin-treated mice, the total circulating nitrates/nitrites (NOx) content in the sera as a biomarker of NO release, 3-nitrotyrosine (3-NT),  Nicotinamide adenine dinucleotide phosphate (NADPH) oxidase 2 (nox2) and NADPH oxidase 4 (nox4) redox-related genes in the aortic tissues as markers of nitro-oxidative stress were assessed [12]. Neither carfilzomib nor metformin or the drugs co-administration led to any significant changes in NOx and 3-NT content (Figure 4A-B), whereas nox2 and nox4 levels did not differ among groups (Figure 4C). Based on these results, endothelial homeostasis is unaffected by carfilzomib and metformin at the selected doses in the sub-acute protocol and sub-acute carfilzomib administration does not lead to nitro-oxidative mediated damage of the aortic tissue.

Concerning the aged mice please refer to page 16; lines 236-244 as follows:

“Carfilzomib increases NOx content in the serum of the aged mice and upregulates iNOS and BIP in the aortic tissue. Co-administration of carfilzomib and metformin leads to an induction of LC3-B-dependent autophagy in the aged murine aortas. 

Given the higher incidence of MM among the elderly population [5], we also assessed the effects of sub-acute carfilzomib and metformin treatment in an aged murine model. As per the data in young mice, carfilzomib did not induce nitro-oxidative damage in the aortas, as demonstrated by the unchanged levels of 3-NT (Figure 5C). However, aged mice exhibited decreased serum NOx levels, and carfilzomib administration increased circulating NOx (Figure 5A-B) which was associated with an upregulation of iNOS in the aortic tissue.”

and in the discussion section please refer to page 30; lines 441-448 as follows:

“Moreover, carfilzomib did not alter NO homeostasis as assessed by circulating NOx levels nor led to nitro-oxidative damage or redox alterations in the aortas, as shown by the stable 3-NT and nox2-nox4 mRNA levels. On the contrary, carfilzomib increased icam-1 mRNA levels, revealing a pro-inflammatory phenotype of the vessels. It has been previously demonstrated that the increased circulating ROS, as observed in our protocol in the Cfz-treated leucocytes (Figure 2G-H), can induce endothelial ICAM-1 expression [35]. Therefore, circulatory oxidative stress can be accredited for icam-1 mRNA upregulation in Cfz-treated aortas. Histological evaluation of the aortas (Figure 3A) did not however provide any evidence for immune cell infiltration of the aortas.”

and for the aged mice please refer to page 31-32; lines 488-492 as follows:

“To increase the translational value of our experiments, we incorporated an aged murine model to investigate the mechanistic effects of carfilzomib on the aged vasculature. In agreement with data from the young mice, carfilzomib did not induce nitro-oxidative damage on the vessels as demonstrated by its surrogate marker 3-NT, despite the increased serum NOx concentration and iNOS expression seen in the carfilzomib-treated aged aortas.”

Comment 4

Figure 3: It was nearly impossible to read the text in the figure (to the Western blot and in immunofluorescence images) due to bad resolution. This may be due to the PDF-conversion, but the text is nevertheless too small. The blot of mTOR phosphorylation is not very representative to the quantitative analysis. mTOR phosphorylation seems to rather increase with Cfz + Met treatment, but this is hard to interpret, as tmTOR bands varies (even within the same samples). If the authors have a more representative blot, I would recommend using it. The same goes for LCB-3, which is more like a smear. Please state in the figure legend what material was used for the blots (whole vessels/aortas).

Reply: We modified Figure 3, so that the western blot bands are more visible and readable. Moreover, we moved the images corresponding to the confocal microscopy of αSMA and LC3-B to Figure S4, in order to make the figure less crowded. Unfortunately, due to time limitations the representative western blot images cannot be modified. Moreover, due to low protein yield of the aortic samples, western blot samples are electrophorized on gradient gels that do not allow the ideal separation of the bands. That’s the reason that LC-3B bands are not perfectly clear.

Concerning the material used for the analysis we have added the respective information in materials and methods section page 35; lines 570-571 as follows: “For histology, cryosectioning and immunoblotting analyses, the thoracic part of the aortas was used by serial sectioning of aortic rings.”

and in page 35; lines 593-594 as follows: “For aortic 3-Nitrotyrosine (3-NT) assessment, the abdominal part of the aortas for the treated animals was snap-frozen, pulverized and extracted with ice cold PBS (0.01M, pH=7.4).”

Comment 5

Figure 4: Representative images for immunofluorescent staining of LC3-B are not shown, only for p53. Why? 4I: The y-axis of the graph states p53/DAPI. In the figure legend, 4I refers to LCB-3/DAPI. Please correct. The y-axis also contains a spelling mistake.

Reply: LC3-B expression was assessed using automated fluorescent microscopy in comparison with p53 that was assessed using conventional confocal microscopy. In order to increase the reliability of our results, the settings and representative cellular images acquired in the automated microscope are presented in Figure S4.

Moreover, we corrected the misspelling of the axis in Figure 4I from “Fuor” to “Fluor” and p53/DAPI to LC3-B/DAPI (New Figure 6 in the revised manuscript).

Comment 6 

In the text to the results of figure 6, the authors state that “comorbidity stimuli led to an increase of p53 expression in HAoSMC”. Looking at the graph in figure 6B, I cannot see any change in the control sample between non stimulated cells and cells stimulated with AngII, CoCl2 or high glucose. Please explain.

Reply: Figure 6B refers to cellular confluency and not to p53/DAPI expression in HAoSMC. As shown in Figure 6 C-D, AngII (100nM) and Glucose (25mM) led to a statistically significant increase in p53/DAPI expression, while CoCl2 led to an increase of the ratio which did not reach statistical significance. Please refer to new Figure 8 in the revised manuscript.

In order to avoid any misinterpretation of the data we corrected the line in page 25; lines 366-367 as follows: “Confirming earlier reports [15, 19-24], AngII (100nM) and Glucose (25mM) led to an increase of p53 expression in HAoSMCs.”

Comment 7

Please write the full term/name (for instance for PGF2a), the first time it is used and not only the abbreviation.

Reply: We included the full term of PGF2α in page 5; line 86.

Comment 8

Materials and method section:

Please add details to the ROS release measurement. What was used to measure ROS release (DCF, L-O12….)?

Please add the source of the human aortic smooth muscle cells.

Reply: We included the needed information about LO-12 in page 35; lines 587-588 as follows: “LO-12 a luminol-based chemiluminescent probe was used for the detection of ROS [57].”

while the source of the human aortic smooth muscle cells (PromoCell) is already stated in the materials and method section in page 38; lines 645-647 as follows: “Primary human aortic smooth muscle cells (HAoSMCs, PromoCell) were seeded in 75ml flasks in optimal Smooth muscle cell growth medium (Smooth muscle cell growth medium kit 2, PromoCell) until they reached confluency.”

Reviewer 2 Report

The study described by Efentakis et al. is an extension of an earlier study by the same group published in 2019.  The earlier study looked at carfilzomib-induced cardiotoxicity in mice, and in the current study, they investigated effects of carfilzomib and metformin on the morphology and physiology of mouse aortic wall cells as well as certain molecular signaling events in the mouse aorta.  They used the carfilzomib toxicity mouse model, which was established in the earlier study, and cultured mouse and human aortic SMCs.  There are several weaknesses.  1. While the cardiotoxicity of carfilzomib is not an acute event, the vascular effects were studies only under acute conditions, and no biological and clinical justifications are provided for this.  It is quite possible that the observed drug-induced vascular alterations are transient and may not result in long-term effects.  In addition, no clinical relevance was given for the manner in which the study was conducted such as the way mice were treated (drug doses, lengths of treatment, route of administration, length of treatment, etc.).  These are important elements when a study is intended for translation. 2. To study the age effect, the authors used cultured cells.  The “young” cells were from mice while the “old” cells were from human.  The effects observed could be due to age difference, to species difference, or both.  The in vitro experiments are not valid for testing age effects.  3. A more valid age effect may be studied using aged mice instead of cultured cells.  I strongly urge the authors to use an aged mouse model to study the age effects.  4. Cultured cells should be used to elucidate molecular signaling events by experimentally interfering (or overactivating) signaling events and showing that vascular events could be inhibited (or ameliorated).   No such attempts were made.  5. Overall, the data presented are all correlative and do not establish a cause-effect relationship of molecular events.  As such, the study appears preliminary in spite of a large volume of data.  Some specific comments are listed below.

  1. Many of the figures are described as “Representative…” such as “Representative graph of body weight.” What do the authors mean by “representative”?  Are there many similar graphs, plots, body weights, cell counts, etc.?
  2. All micrographs are of poor quality. Images do not have sufficient resolution and they appear fussy.  There is little value as data, and one cannot trust biological conclusions based on these images.
  3. When the exact number of “n” is given, sometimes, it does not correspond with the points indicated in the figure (for example, Fig. 4A shows 4 data points, not 3).
  4. Lines237-238. The authors state, “in contrast to the prmVSMCs, the compounds did not reduce significantly MTT conversion in the HAoSMCs, whilst metformin at the low dose led to an increased MTT metabolic conversion.”  This could be due to cell senescence, species difference, or a combination of both.  These and other results obtained from cultured cells cannot be interpreted in a straightforward manner.
  5. Although Figure 7 is very attractive, this is an over-interpretation of results that do not establish the cause-effect relationship of these molecular events. Much more work is necessary to justify publication of this figure.
  6. Animals. Describe the method of euthanasia used.
  7. Aorta. When doing various analyses of mouse aortas, were the same region of the aorta used for each analysis?  For example, which area of the aorta was used to do the relaxation and contraction experiment, histology, immunoblotting, cryosectioning, etc.?  Consistent sampling is critical because different parts of the aorta are developmentally different.  There were only 5 aortas for each group.  Were all of them used to do the physiological, morphological, immunohistochemcial, and biochemical analyses?  Please describe which region was used for each experiment.
  8. There is no description as to how leucocyte ROS was measured. The method only describes how cells were stimulated.
  9. For Western blotting, the authors made aortic tissue powder. Why?  Describe how this was done.  Also, give reasons why the authors chose so many antibodies for Western blotting?  The authors state, “PVDF membranes were incubated with secondary antibodies for 2 h at 24 ° C [goat anti-rabbit HRP (# 7074); Cell Signaling Technology, Beverly, MA, USA].”  Were all primary antibodies made in rabbits?
  10. English editing is highly receommended.

Author Response

We would like to thank the Editors and Reviewers for evaluating our manuscript. During the revision process, we have addressed all comments raised point by point which undoubtedly provided us with a valuable opportunity to improve our manuscript. We hope that the revised manuscript will be acceptable for publication and will be of interest to the readers of International Journal of Molecular Sciences.

We have marked the modified texts as highlighted yellow in the revised manuscript.

Reviewer 2

The study described by Efentakis et al. is an extension of an earlier study by the same group published in 2019.  The earlier study looked at carfilzomib-induced cardiotoxicity in mice, and in the current study, they investigated effects of carfilzomib and metformin on the morphology and physiology of mouse aortic wall cells as well as certain molecular signaling events in the mouse aorta.  They used the carfilzomib toxicity mouse model, which was established in the earlier study, and cultured mouse and human aortic SMCs.  There are several weaknesses.

Reply: We would like to thank the reviewer for their substantial comments, which were really helpful in improving the manuscript.

Comment 1

While the cardiotoxicity of carfilzomib is not an acute event, the vascular effects were studies only under acute conditions, and no biological and clinical justifications are provided for this.  It is quite possible that the observed drug-induced vascular alterations are transient and may not result in long-term effects.  In addition, no clinical relevance was given for the manner in which the study was conducted such as the way mice were treated (drug doses, lengths of treatment, route of administration, length of treatment, etc.).  These are important elements when a study is intended for translation.

Reply: We thank very much the reviewer for raising these important points. Carfilzomib’s induced cardiomyopathy is well-described to be induced acutely, presented as acute heart failure in treated patients. However, it is noted that the induced cardiotoxicity is reversible upon discontinuation of the medication (Dimopoulos MA, et al. Blood Adv. 2017; 1(7): 449–454). In our previous research work, addressing the cardiotoxicity of carfilzomib in C57Bl/6 mice, we investigated the carfilzomib-induced cardiac dysfunction, by incorporating different timepoints of experimentation. We found that four doses of carfilzomib administration lead to an established cardiac dysfunction. Upon discontinuation of the therapy, cardiac function of the mice was restored to baseline levels. The latter findings were in compliance with the clinically observed data and therefore our protocol is of high translational value. Moreover, the selected doses and dose regimen are in complete compliance with the clinically applied regimens and were calculated in detail using interspecies formulas for dose calculations. Therefore, it is already established that by both a functional and a mechanistic aspect the sub-acute carfilzomib protocol can successfully recapitulate the clinical events manifested in treated patients (Efentakis P et al. Blood 2019, 133, (7), 710-723).

In the revised manuscript we added the following sentence in page 34; lines 563-566 as follows: “Drug doses, treatment-duration and route of administration were established in our previous study addressing the cardiotoxicity of carfilzomib; therefore, is already established that by both a functional and a mechanistic approach the 4-dose carfilzomib administration regimen can appropriately recapitulate the clinical events manifested in treated patients [11].”

Concerning the vascular effects and spasmogenicity of carfilzomib, vascular adverse effects also appear in an acute but transient manner in multiple myeloma patients, as presented in a case study report (KocabaÅŸ U et al. Indian J Hematol Blood Transfus. 2019, 1-2; discussed in page 29; lines 429-431). Therefore, our protocol is accredited with high translational value, concerning the drug doses, lengths of treatment and route of administration.      

Comment 2

To study the age effect, the authors used cultured cells.  The “young” cells were from mice while the “old” cells were from human.  The effects observed could be due to age difference, to species difference, or both.  The in vitro experiments are not valid for testing age effects. A more valid age effect may be studied using aged mice instead of cultured cells.  I strongly urge the authors to use an aged mouse model to study the age effects.

Reply: The Reviewer is absolutely right about the discrepancies that might occur due to age difference, to species difference, or both and we thank them very much for raising this important point. To resolve this, we included additional in vivo experiments in order to connect the animal with the human in vitro data. To facilitate this, we recapitulated the experiments in aged C57Bl/6 mice (15-17 months of age) which is established as an aging in vivo model. Co-administration of the drugs led to a synergistic induction of LC3-B-dependent autophagy, which is in line with our in vivo data from the young mice and our in vitro data.

Please refer to the revised manuscript in the results section in page 16; lines 236-255 and in Figure 5 as follows:  “Carfilzomib increases NOx content in the serum of the aged mice and upregulates iNOS and BIP in the aortic tissue. Co-administration of carfilzomib and metformin leads to an induction of LC-3B-dependent autophagy in the aged murine aortas. 

Given the higher incidence of MM among the elderly population [5], we also assessed the effects of sub-acute carfilzomib and metformin treatment in an aged murine model. As per the data in young mice, carfilzomib did not induce nitro-oxidative damage in the aortas, as demonstrated by the unchanged levels of 3-NT (Figure 5C). However, aged mice exhibited decreased serum NOx levels, and carfilzomib administration increased circulating NOx (Figure 5A-B) which was associated with an upregulation of iNOS in the aortic tissue. In agreement with the mechanistic data from the young murine aortas, carfilzomib upregulated BIP, a marker of unfolded protein response. Carfilzomib and the co-administration of carfilzomib and metformin did not affect Raptor phosphorylation or expression but led to an increase in mTOR inhibitory phosphorylation at S2481. Metformin increased eNOS phosphorylation compared to the control group, and metformin as well as the drug co-administration increased both eNOS and AMPKα phosphorylation compared to carfilzomib alone. Furthermore, metformin and Cfz+Met led to increased LC3-B expression (Figure 5D-E). Importantly and in compliance with the young mice data, the drugs’ co-administration led to a decrease in αSMA.

Collectively, the mechanistic data obtained from both the young and the aged murine models show that sub-acute co-administration of carfilzomib and metformin induces a LC3-B-dependent autophagy in the vessels which is independent of age. These effects are observed mostly in the vascular smooth muscle cell layer of the aortas.” 

and in the discussion section in pages 31-32; lines 488-502 as follows: “To increase the translational value of our experiments, we incorporated an aged murine model to investigate the mechanistic effects of carfilzomib on the aged vasculature. In agreement with data from the young mice, carfilzomib did not induce nitro-oxidative damage on the vessels as demonstrated by its surrogate marker 3-NT, despite the increased serum NOx concentration and iNOS expression seen in the carfilzomib-treated aged aortas. Mechanistically, metformin increased AMPKα and eNOS phosphorylation -independently of carfilzomib treatment- an effect that is already shown to be vasoprotective in aged vessels [45]. Carfilzomib and metformin co-administration led to increased AMPKα and eNOS phosphorylation, as well as to a synergistic induction of autophagy as shown by mTOR inhibitory phosphorylation at S2481, and increased LC3-B expression. Moreover, a significant decrease of αSMA – a surrogate marker of the VSMCs- was observed only in the Cfz+Met group. The induction of autophagy in aged vessels ameliorates vascular hypertrophy and exerts vasoprotective effects, by decreasing αSMA expression as a result of decreased VSMCs proliferation [46]. Therefore, carfilzomib and metformin co-administration exerts a significant vasoprotective effect on the aged vessels via increased vascular LC3-B-dependent autophagy and increased eNOS and AMPKα phosphorylation.”

Comment 3

Cultured cells should be used to elucidate molecular signaling events by experimentally interfering (or overactivating) signaling events and showing that vascular events could be inhibited (or ameliorated).   No such attempts were made.

Reply: We thank the reviewer for their suggestion. Unfortunately, due to the time limitations and the already high volume of data this cannot be addressed at the moment. However, the latter concerns will be addressed in a future research work.

We have included the last concern in the end of the discussion section in page 34; lines 545-547 as follows: “Moreover, additional in vitro experiments aiming to elucidate molecular signaling events by experimental interfering are also needed, to fully elucidate the effects of carfilzomib and metformin on the vascular musculature.”

Comment 4

Overall, the data presented are all correlative and do not establish a cause-effect relationship of molecular events.  As such, the study appears preliminary in spite of a large volume of data.  Some specific comments are listed below. Many of the figures are described as “Representative…” such as “Representative graph of body weight.” What do the authors mean by “representative”?  Are there many similar graphs, plots, body weights, cell counts, etc.?

Reply: In the current research work we have performed a laborious work investigating the molecular cascades induced by carfilzomib and metformin as well as by the co-administration of the drugs. Complete molecular pathways regarding the autophagy (i.e. mTOR, Raptor, AMPKα, Beclin-1, LC3-B), AMPKα (ACC, AMPKα), NO regulation (eNOS, iNOS, VASP, SRC) and Akt signaling cascades have been investigated. Therefore, the molecular cascades that can be involved in the observed phenotype are extensively studied. Moreover, we have conducted additional experiments, in order to investigate surrogate markers of NO homeostasis (serum NOx) and nitro-oxidative stress (3-Nitrotyrosine) in order to verify the findings from the immunoblotting analysis in both young and aged mice.

Concerning the possible metabolic effects of the drugs, we have conducted a complete in vitro analysis of the cellular metabolism using various mitochondrial complex inhibitors in both young and aged vascular smooth muscle cells. Therefore, the effect of the drugs on mitochondrial respiration and anaerobic glycolysis is completely investigated in vitro.

Conclusively, the current work provides a clarified representation of molecular and metabolic effects of the drugs, allowing safe conclusion regarding the effect of carfilzomib and metformin on the young and aged vasculature.

We corrected the statement representative in the figure legends.

Comment 5

All micrographs are of poor quality. Images do not have sufficient resolution and they appear fussy.  There is little value as data, and one cannot trust biological conclusions based on these images.

Reply: We have improved the quality of all figures as much as possible according to the instructions of the Journal. Please see the revised figures in the revised manuscript. 

Comment 6

When the exact number of “n” is given, sometimes, it does not correspond with the points indicated in the figure (for example, Fig. 4A shows 4 data points, not 3).

Reply: We thank the reviewer for this point and we apologized for the typographical mistake. We have corrected the figure legend in Figures 4, 5 and 6A respectively (New Figures 5, 7 and 8 in the revised manuscript).

Comment 7

Lines237-238. The authors state, “in contrast to the prmVSMCs, the compounds did not reduce significantly MTT conversion in the HAoSMCs, whilst metformin at the low dose led to an increased MTT metabolic conversion.”  This could be due to cell senescence, species difference, or a combination of both.  These and other results obtained from cultured cells cannot be interpreted in a straightforward manner.

Reply: We agree with reviewer that as in vitro experiments lack the complexity of the in vivo experiments and can be affected by cell senescence, species difference, or a combination of both. To resolve this concern, we have incorporated the in vivo aged model (please see answer in Comment 2).

The concerns about the in vitro experiments are added in the in the end of the discussion section in page 33; lines 541-542 as follows: “In vitro experiments do not fully replicate the complexity of the pathological mechanisms evident in animal models and/or in humans.”  

Comment 8

Although Figure 7 is very attractive, this is an over-interpretation of results that do not establish the cause-effect relationship of these molecular events. Much more work is necessary to justify publication of this figure.

Reply: We thank very much the reviewer and we have revised Figure 7 according to reviewer’s instructions. Please refer to Figure 9 in the revised manuscript.

Comment 9

Animals. Describe the method of euthanasia used.

Reply: We have included the method of euthanasia in the materials and methods section in page 34; lines 566-568, described as: “At the end of the experiments, animals were anaesthetized by ketamine (100mg/kg) and were euthanized by cervical dislocation.”

Comment 10

Aorta. When doing various analyses of mouse aortas, were the same region of the aorta used for each analysis?  For example, which area of the aorta was used to do the relaxation and contraction experiment, histology, immunoblotting, cryosectioning, etc.?  Consistent sampling is critical because different parts of the aorta are developmentally different.  There were only 5 aortas for each group.  Were all of them used to do the physiological, morphological, immunohistochemcial, and biochemical analyses?  Please describe which region was used for each experiment.

Reply: All experiments, in both young and aged C57Bl/6 mice were conduced by the use of the thoracic part of the aorta. One ring (3mm) just after the aortic arch was used for the relaxation and contraction experiment, one ring (3mm) was used for the cryosectioning, one ring (5mm) for the histology while the rest of the thoracic part of the aorta was used for the immunoblotting analysis. 3-Nitrotyrosine was assessed in the abdominal part of the aortas due to shortage of biomaterial for the analyses.

In order to resolve this concern, we have added the following section in the materials and methods section in page 35; lines 570-571 as follows: “For histology, cryosectioning and immunoblotting analyses, the thoracic part of the aortas was used by serial sectioning of aortic rings.”

and in page 35; lines 593-594 described as: “For aortic 3-Nitrotyrosine (3-NT) assessment, the abdominal part of the aortas for the treated animals was snap-frozen, pulverized and extracted with ice cold PBS (0.01M, pH=7.4).”

Comment 11

There is no description as to how leucocyte ROS was measured. The method only describes how cells were stimulated.

Reply: We included the needed information about LO-12 in page 35; lines 587-588 as follows:  “LO-12 a luminol-based chemiluminescent probe was used for the detection of ROS [57].”

Comment 12

For Western blotting, the authors made aortic tissue powder. Why?  Describe how this was done.  Also, give reasons why the authors chose so many antibodies for Western blotting?  The authors state, “PVDF membranes were incubated with secondary antibodies for 2 h at 24 ° C [goat anti-rabbit HRP (# 7074); Cell Signaling Technology, Beverly, MA, USA].”  Were all primary antibodies made in rabbits?

Reply: The pulverization of the aorta was performed in a mortar using a small pestle, in liquid nitrogen on dry ice. The latter information is added in the materials and methods section in page 36; lines 606-609 as follows: “Thoracic aortic tissue was pulverized in a mortar in liquid nitrogen on dry ice and aortic powder was extracted with lysis buffer (1% Triton X-100, 20 mM Tris pH 7.4-7.6, 150 mM NaCl, 50 mM NaF, 1 mM EDTA, 1 mM EGTA, 1 mM glycerophosphatase, 1% SDS, 100 mM PMSF and 0.1% phosphatase-protease inhibitors cocktail).”

 The aforementioned method of tissue homogenization was selected as aortic tissue is a very elastic tissue, therefore, other methods of tissue dissociation result in very low protein yield (i.e by using metal beads).

The multiple protein targets were selected as carfilzomib’s effects on the vasculature could be multifaceted. Therefore, AMPKα signaling (i.e. AMPKα and ACC), NO-implicated proteins (i. e. eNOS, VASP, SRC), autophagy-associated molecules (i.e. mTOR, Raptor, LC3-B and beclin) and antiapoptotic cascades (i. e. Akt) were investigated.

As indicated in the manuscript all antibodies were raised in rabbit and are Rabbit IgG, so that they won’t non-specifically bind to the mouse IgGs in the samples.

Comment 13

English editing is highly recommended.

Reply: We have carefully revised the manuscript so that linguistic or grammar deficits have been corrected.